# Aligning Agent Policy with Principal Interests:
# Reward Design via Bilevel RL

**Souradip Chakraborty** [1]  **Amrit Singh Bedi** [1]  **Alec Koppel** [2]  **Dinesh Manocha** [1]  **Furong Huang** [1]  **Mengdi Wang** [3]

## Abstract

In reinforcement learning (RL), a reward function is often assumed at the outset of a policy optimization procedure. Learning in such a fixed reward paradigm in RL can neglect important policy optimization considerations, such as state space coverage and safety. Moreover, it can fail to encompass broader impacts in terms of social welfare, sustainability, or market stability, potentially leading to undesirable emergent behavior and potentially misaligned policy. To mathematically encapsulate the problem of aligning RL policy optimization with such externalities, we consider a bilevel optimization problem and connect it to a principal-agent framework, where the principal specifies the broader goals and constraints of the system at the upper level and the agent solves a Markov Decision Process (MDP) at the lower level. The upper-level deals with learning a suitable reward parametrization corresponding to the broader goals and the lower-level deals with learning the policy for the agent. We propose Principal driven Policy Alignment via Bilevel RL (PPA-BRL), which efficiently aligns the policy of the agent with the principal's goals. We explicitly analyzed the dependence of the principal's trajectory on the lower-level policy, prove the convergence of PPA-BRL to the stationary point of the problem. We illuminate the merits of this framework in view of alignment with several examples spanning energy-efficient manipulation tasks, social welfare-based tax design, and cost-effective robotic navigation.

*Equal contribution [1]Department of Computer Science, University of Maryland, College Park, USA. [2]JP Morgan Chase AI Research, USA. [3]Department of Electrical Engineering, Princeton University/Deepmind, Princeton, NJ, USA. Correspondence to: Souradip Chakraborty <schakra3@umd.edu>.

*Interactive Learning with Implicit Human Feedback Workshop at ICML 2023*, Honolulu, Hawaii, USA. PMLR 202, 2023. Copyright 2023 by the author(s).

## 1. Introduction

The increasing complexity and widespread use of artificial agents highlight the critical need to ensure that their performance aligns with broader objectives such as social welfare and economic impacts, which are referred to by economists as externalities (Rahwan, 2018). This has given rise to the field of AI alignment (Liu et al., 2022), which focuses on aligning agent behavior with societal goals and values. We study this problem in the context of reinforcement learning (RL) (Sutton & Barto, 1998), which may be mathematically posed as a Markov Decision Problem (Puterman, 2014). This framework gained traction for its ability to solve games (Mnih et al., 2013; Silver et al., 2017), personalized recommendation systems (Steck et al., 2021), various continuous control tasks (Ng et al., 1999; Lillicrap et al., 2015), and develop strategies in financial markets (Ardon et al., 2021). The question of how to align an RL agent to broader goals may be decomposed into two major questions:

(i) *How can one align the behavior of learning agents with broader objectives?*

(ii) *Does alignment compromise the performance of learning agents?*

To address (i), a litany of approaches have been studied: inverse reinforcement learning (Ziebart et al., 2008; Brown et al., 2019; Arora & Doshi, 2020) and imitation learning (Ho & Ermon, 2016; Kang et al., 2018; Ghasemipour et al., 2019; Xiao et al., 2019) employ expert demonstrations to learn a policy or reward function (Torabi et al., 2018; Ouyang et al., 2022). Offline approaches leverage large-scale prior-collected data to learn behaviors deemed appropriate by an expert (Yang et al., 2023; Levine et al., 2020; Yin et al., 2022; Chen et al., 2022). However, acquiring a substantial amount of human feedback or well-aligned demonstrations can be expensive or infeasible (Bai et al., 2022; Chen et al., 2023a; Wolf et al., 2023), especially when dealing with diverse (Jabbari et al., 2017), personalized, or competing goals set by individuals, organizations, or governments (Bai et al., 2022; Chen et al., 2023a; Liu et al., 2022; Sun et al., 2023). Therefore, instead, we adopt an approach where one seeks to directly design the data-

generating mechanism according to some external utility function. Doing so introduces a dependency on the question (ii), as alignment upends the standard training mechanism for learning agents. In particular, evaluating an alignment objective is contingent upon the agent's performance over a time horizon through trajectories generated by the agent's policy. This entanglement naturally leads to a variant of bilevel optimization.

To date, problems of this type have mostly been studied through the lens of bilevel optimization (Bracken & McGill, 1973). Bilevel optimization considers a setup where the outer-level objective is a function of decision variables and the optimizer at the inner-level in both deterministic (Hansen et al., 1992; Shi et al., 2005) or a probabilistic sense (Chen et al., 2021). However, these works primarily do not allow either stage to be defined by an MDP.

Distinct lines of effort exist from game theory and economics in terms of mechanism design (Myerson, 1989; Hurwicz, 2003; Maskin, 2008), Stackleberg games (Von Stackelberg, 2010), and contract theory (Green & Stokey, 1983; Eisenhardt, 1989; Stiglitz, 1989) that have a bearing on our technical approach. Closer to this work is recent intersections of MDPs and algorithmic Stackleberg solvers, which impose special information structures such as linear MDPs (Zhong et al., 2021) or oracle access to transition models (Goktas et al., 2022). Most similar to our work are those that develop implicit function-theorem based gradient play (Fiez et al., 2020; Vu et al.); however, these techniques are experimentally-based. Worth mentioning is principal-agent problem in contract theory, which admits a Stacklberg formulation (Zhu et al., 2022; Chen et al., 2023b) in terms of repeatedly interacting bandits.

A more extensive discussion of these research threads is available in Appendix 8. The main takeaway is that no formulation exists in which the inner-stage is a parameterized policy optimization problem in an MDP, and the outer-stage is an expected-value objective over the distribution underlying trajectories defined by an RL policy. This setting, to our knowledge, is most appropriate for formalizing the alignment of RL agents to externalities (Rahwan, 2018) and emergent behavior (Teo et al., 2013).

This generality introduces technical impediments not found in either prior RL methods or bilevel programming: (i) the outer objective is parameterized by the inner-stage's optimal policy, which must be numerically approximated; (ii) to solve the aforementioned numerical approximation, the local Karush-Kuhn-Tucker (KKT) conditions of the outer stage exhibit dependence on Jacobians and Hessians that are interrelated with variables at the inner-stage (Hong et al., 2020; Khanduri et al., 2021; Li et al., 2022); and (iii) the sampling distribution for the expectation at the outer-stage depends on trajectories generated from product measure as-

sociated with the transition dynamics conditioned on fixed policy at the inner-stage. We develop an algorithmic framework that addresses these issues and therefore enables one to design RL agents that are aligned with externalities. Therefore, our main **contributions** are to:

- formulate the agent policy alignment problem as a bilevel optimization problem where the outer objective centers on reward design through policy evaluation over the horizon, and the inner level pertains to policy alignment with the designed reward via policy optimization in an MDP;

- by analyzing local KKT points using differentiation and the Implicit function theorem, we derive an iterative procedure to jointly solve for the design parameters at the outer level and policy parameters at the inner level in this framework. This procedure, known as Principal-driven Policy Alignment via Bilevel RL (PPA-BRL);

- establish the proposed methodology converges to a local KKT point of the problem. The main challenge lies in deriving and dealing with the expectation of the gradient of the outer objective, which depends on the policy of the inner level. This results in an interesting notion of score function with respect to outer parameters crucial for design and alignment.

## 2. Policy Alignment as Bilevel Formulation

**Standard policy optimization.** Let us start by considering the Markov Decision Process (MDP) tuple $\mathcal{M} := \{\mathcal{S}, \mathcal{A}, \gamma, \mathbb{P}, r\}$ (Puterman, 2014), which is a tuple consisting of a state space $\mathcal{S}$, action space $\mathcal{A}$, transition dynamics $\mathbb{P}$, discount factor $\gamma \in (0, 1)$, and reward $r : \mathcal{S} \times \mathcal{A} \to \mathbb{R}$. Starting from a given state $s \in \mathcal{S}$, an agent selects an action $a$ and transitions to another $s' \sim \mathbb{P}(\cdot \mid s, a)$. We hypothesize that the agent follows a stochastic stationary policy that maps states to distributions over actions $\pi_\theta : \mathcal{S} \to \mathbb{P}(\mathcal{A})$, which is parameterized by a parameter vector $\theta \in \mathbb{R}^d$. We can write the standard finite horizon policy optimization problem as

$$\max_\theta V_s(\theta) := \mathbb{E}\left[\sum_{h=0}^{H-1} \gamma^h r(s_h, a_h) \mid, s_0 = s\right], \quad (1)$$

where the expectation is with respect to the stochasticity in the policy $\pi_\theta$ and the transition dynamics $\mathbb{P}$. In (1), we used notation $V_s(\theta)$ for value function in state $s$ to emphasize the dependence on parameters $\theta$. We note that the formulation in (1) utilizes a reward function $r$ (fixed a priori), and we learn a policy corresponding to reward function $r$. But as detailed in the introduction, policy learning with a fixed reward does not allow one to tether the training process

to an external objective such as (Ni & Paul, 2019), social welfare (Balcan et al., 2014), or market stability (Buehler et al., 2019).

**Proposed formulation.** To enable this capability, we formulate a bilevel optimization problem where the inner level is a Markov Decision Process (MDP) and The outer level is an expected utility maximization, whose distribution depends upon the transition model and policy associated with trajectories at the inner level:

$$\max_{\nu} \quad G(\nu, \theta^*(\nu)) \tag{2}$$

$$\text{s.t. } \theta^*(\nu) := \arg\max_{\theta} \mathbb{E}\left[\sum_{h=0}^{H_\ell - 1} \gamma^h r_\nu(s_h, a_h) \mid s_0 = s\right],$$

where $a_h \sim \pi_\theta(a_h|s_h)$ and the inner-level goal is to maximize the $H_\ell$ horizon $\gamma$-discounted cumulative return of rewards $r_\nu(s, a)$, i.e., the value function.

**Inner Objective.** As previously mentioned, the interrelationship of these levels is intended to formalize that a designer, called a *principal* in contract theory (Zhu et al., 2022; Zhao et al., 2023), specifies the incentive structure associated with the reward function of an RL agent at the inner-level. To be more precise, note that the reward $r_\nu(s, a)$ in (2) is additionally a parametric function of the designer's parameters $\nu \in \mathbb{R}^n$, in contrast to the standard MDP setting (1). Hence, we can also write the inner-level optimization problem in (2) as

$$\max_{\theta} V_s(\nu, \theta) := \mathbb{E}\left[\sum_{h=0}^{H_\ell - 1} \gamma^h r_\nu(s_h, a_h) \mid s_0 = s\right], \tag{3}$$

where $a_h \sim \pi_\theta(a_h|s_h)$ and the expectation is with respect to the stochasticity in the policy $\pi_\theta$ and the transition dynamics $\mathbb{P}$. In contrast to value function in (1), the value function in (3) is an implicit function of the designer's parameters $\nu$ which controls the reward function $r_\nu$. For the example of robotic navigation, $r_\nu(s, a)$ is a function of the distance to a goal location. Further, to make the policy parameter $\theta$ dependence on the reward parameter $\nu$ clear, we write $\theta(\nu)$ subsequently. In this work, we restrict focus to the case that the optimizer $\theta^*(\nu)$ at the inner-level is unique, which mandates that one parameterize the policy in a tabular (Agarwal et al., 2020; Bhandari & Russo, 2021) or softmax fashion (Mei et al., 2020a); otherwise, at most one can hope for with a policy gradient iteration is to obtain an approximate local extrema (Zhang et al., 2020). We defer a more technical discussion of this aspect to Section 4. For the upper-level objective in (2), we aim to maximize an objective function $G(\nu, \theta^*(\nu))$, which depends on the design parameters $\nu$ and a cumulative return of the RL agent evaluated at a given policy, which is an implicit function of policy parameters $\theta^*(\nu)$. The outer objective in (2) addresses the limitations of (1)

encompasses the goal of aligning the reward function with objectives such as energy usage (Ni & Paul, 2019), social welfare (Balcan et al., 2014), market stability (Buehler et al., 2019), or other external factors not under the direct control of an RL agent. The parameters $\nu$ under the designer's control define the incentive structure for the behavior of an RL agent, and in that way, exhibits parallel with mechanism design (Lyu et al., 2022a) and contract theory (Zhu et al., 2022).

**Outer objective.** To be specific, we consider a utility at the outer level of the form

$$G(\nu, \theta^*(\nu)) = D(\pi_{\theta^*(\nu)}) + Z(\nu), \tag{4}$$

which is comprised of two terms: a quantifier $U(\pi_{\theta^*(\nu)})$ of the merit of design parameters $\nu \in \mathbb{R}^n$, whereas the second term $Z(\nu)$ represents, e.g., a regularizer or prior on the distribution over trajectories $\mathbb{P}(\tau; \theta^*(\nu))$. More specifically, the explicit mathematical form of $U(\pi_{\theta^*(\nu)})$ decides the quality of policy $\pi_{\theta^*(\nu)}$ by collecting trajectories (denoted by $\tau$), and via associating a designer's reward $U(\tau)$, for each trajectory under policy $\pi_{\theta^*(\nu)}$ given by

$$D(\pi_{\theta^*(\nu)}) = \mathbb{E}_{P(\tau; \theta^*(\nu))}[U(\tau)] = \sum_{\tau} U(\tau) \cdot \mathbb{P}(\tau; \theta^*(\nu)), \tag{5}$$

where $\mathbb{P}(\tau; \theta^*(\nu))$ denotes the probability of trajectory $\tau$. Subsequently, we employ the shorthand notation $\mathbb{E}_{\pi_{\theta^*(\nu)}}[\cdot]$ to denote the expectation with respect to the distribution underlying trajectory $\tau$: $\prod_{h=0}^{H_u} \mathbb{P}(s_{h+1} \mid s_h, a_h)\pi_{\theta^*(\nu)}(\cdot|s_h)$, where $H_u$ is the length of upper level trajectory collected at the lower level optimal policy. The explicit form of such an objective, i.e., reward specifications for the designer $U$, is provided in detailed examples in the following Subsection 2.1.

As we will see in the next section, developing an iterative solver to (2) exhibits some unique technical challenges not found in prior art on bilevel programming (Ghadimi & Wang, 2018a; Akhtar et al., 2022). Before shifting focus to do so, we present some representative examples of (2).

## 2.1. Motivating Examples

**Example 1: Energy efficient and sustainable design for robotic manipulation.** Consider a robotic manipulation task where the objective of the agent is to learn an optimal policy to transport components from a fixed position to a target position $\nu := (x, y)$. On the other hand, the designer's objective is to select the work-bench position $\nu$ to minimize the energy consumption of the robotic arm during the transportation task. Hence, it naturally boils down to the

following bilevel problem as

$$\max_{\nu:=(x,y)} \mathbb{E}_{P(\tau;\theta^*(\nu))} \left[ \sum_{h=1}^{H_u} -a_h w_h \mid a_h \sim \pi_{\theta^*(\nu)}(\cdot|s_h) \right] \tag{6}$$

$$\text{s.t. } \theta^*(\nu) := \arg\max_\theta \mathbb{E} \left[ \sum_{h=0}^{H_\ell-1} \gamma^h r_\nu(s_h, a_h) \mid, s_0 = s \right],$$

where $a_h \sim \pi_\theta(a_h|s_h)$, $\mathbb{E}_{\pi_{\theta^*(\nu)}}$ denotes the expectation with respect to the trajectories collected by the lower-level optimal policy $\pi_{\theta^*(\nu)}$. In (6), action $a_h$ denotes the angular acceleration of the robotic arm, the state is represented by $s_h = (\alpha_h, w_h)$, $\alpha_t$ is the discretized angle, and $w_h$ angular velocity of the robotic arm. we define the transitions as $(\alpha_{t+1}, w_{t+1}) = (\alpha_h + w_h, w_h + a_h)$. The reward of the inner objective $r_\nu(s_h, a_h) = -\lambda_1 \|s_h - \nu\|^2 - \lambda_2 \|w_h\|^2$, i.e., reward increases as the arm moves closer to the workbench with a controlled angular velocity. The outer objective focuses on minimizing the energy emission during transport and is thus entangled with the trajectories generated under the optimal policy obtained via the lower-level.

**Example 2: Social welfare aligned tax design for households.** Consider the problem of tax design for individual households while remaining attuned to social welfare, motivated by (Hill et al., 2021). From the household's perspective, each household seeks to maximize its own utility $u_h$ based on the number of working hours, consumption of goods, and net worth. Let us denote the accumulated asset as state $s_h$. At each time step $h$, the household agent selects an action $a_h = (n_h, c_{i,h})$, $a_h \sim \pi_\theta(a_h|s_h)$, where $n_h$ is the number of hours worked, and $c_{i,h}$ is the consumption of good $i$ at a pre-tax price of $p_i$, and $\theta$ denotes the policy parameter. We denote $f(s_h)$ as the reward for the accumulative asset $s_h$, updated at each time step by $s_{h+1} = s_h + (1-x)wn_h - \sum_{i=1}^M c_{i,h}$ and $\nu = (x, y_i)$ is the income tax rate and consumption tax rate for good $i$. Here we note that $x$ is a uniform tax across all households, whereas $y_i$ is a household-specific tax rate. Then the household agent's utility at time step $h$ is given by the equation $u_h = f(s_h) - \gamma n_h^2 + \prod_{i=1}^M \left( \frac{c_{i,h}}{p_i(1+y_i)} \right)^{\nu_i}$, where the product term corresponds to Cobb-Douglas function (Roth et al., 2016). In contrast, the objective of the regulatory body or government (upper-level) is to maximize the social welfare $v_t$ by adjusting the tax rates $\nu$ based on the optimal policy of the household agent (lower level). Hence, the outer objective representing the social welfare is defined as $v_h = g(s_h) + \sum_{i=1}^M \frac{c_{i,h}}{1+y_i} + \psi \ln \left( \frac{\prod_{i=1}^M c_{i,t}y_i}{\prod_{i=1}^M (1+y_i)} + wx_n h \right)$, where $g(\cdot)$ is the reward for the accumulative asset, $\psi$ is a positive constant, and $w$ is the wage rate. The household agent follows a policy that maximizes its discounted cumulative reward, while the social planner aims to maximize the discounted total social welfare by tuning the tax rates $x$ and $y_i$. Thus, the bilevel formulation is given by

$$\max_\nu \mathbb{E}_{P(\tau;\theta^*(\nu))} \left[ \sum_{h=0}^{H_u-1} v_\nu(s_h, a_h) \right] \tag{7}$$

$$\text{s.t. } \theta^*(\nu) := \arg\max_\theta \mathbb{E} \left[ \sum_{h=0}^{H_\ell-1} \gamma^h u_\nu(s_h, a_h) \mid, s_0 = s \right].$$

where $a_h \sim \pi_\theta(a_h|s_h)$, $\gamma$ is the discount rate, and $\theta^*(\nu)$ represents the optimal inner policy of the household agent, which maximizes its expected cumulative return over a time horizon $H_\ell$.

## 3. Algorithmic Policy Alignment with Externalities via Bilevel RL

To solve the bilevel problem in (2), we seek to develop stochastic optimization algorithm similar to (Ghadimi & Wang, 2018a) which requires to evaluate the gradients of the upper and lower level objective. To do so, we begin by deriving the gradient of the outer objective $\nabla_\nu G[\nu, \theta^*(\nu)]$ from (4) with respect to design parameter $\nu$ given by

$$\nabla_\nu G[\nu, \theta^*(\nu)] = \nabla_\nu \sum_\tau U(\tau) \cdot \mathbb{P}(\tau; \theta^*(\nu)) + \nabla_\nu Z(\nu)$$

$$= \mathbb{E}_\tau[U(\tau) \cdot \nabla_\nu \log(\mathbb{P}(\tau; \theta^*(\nu)))] + \nabla_\nu Z(\nu)$$

$$= \mathbb{E}_{P(\tau;\theta^*(\nu))} \left[ U(\tau) \cdot \sum_{h=0}^{H_u-1} \nabla_\nu \log \pi_{\theta^*(\nu)}(a_h|s_h) \right]$$

$$+ \nabla_\nu Z(\nu), \tag{8}$$

where we have used the log-trick and standard rule of expectation to get the final expression in (8), similar to the standard derivation of the policy gradient formula (Agarwal et al., 2020), and the score function method in general (Williams, 1992; Sutton et al., 1999). We emphasize two aspects: (a) the *score function* term $\nabla_\nu \log \pi_{\theta^*(\nu)}(a|s)$ in (8), which denotes the gradient of the optimal policy with respect to the design parameter $\nu$; and (b) the expectation is with respect to the trajectory distribution $\mathbb{P}(\tau; \theta^*(\nu))$ generated under policy at the inner-level $\pi_{\theta^*(\nu)}$ given by $\mathbb{P}(\tau; \theta^*(\nu)) := \rho(s_0) \prod_{h=0}^{H_\ell-1} \pi_{\theta^*(\nu)}(a_h|s_h) \mathbb{P}(s_h' \mid \pi_{\theta^*(\nu)}(s_h, a_h))$.

True to our knowledge, this is the first time this *coupled* score function is appearing in the RL training, which captures the change of optimal policy with respect to the reward function's design parameters. The term $\nabla_\nu \log \pi_{\theta^*(\nu)}(a|s)$ is crucial for our setting, as the designer (principal agent, such as a regulatory body or central planner) at the upper-level can directly control the policy learning by modifying the reward parameters. However, the estimation of $\nabla_\nu \log \pi_{\theta^*(\nu)}(a_h|s_h)$ is nontrivial as it depends on the solution of the lower-level problem in (2), and therefore requires the evaluation of hypergradient $\nabla_\nu \theta^*(\nu)$. To see

that, let us employ the shorthand notation $f_h(\theta^*(\nu)) := \log \pi_{\theta^*(\nu)}(a_h|s_h)$, we can rewrite the gradient as

$$\nabla_\nu f_h(\theta^*(\nu)) = \nabla_\nu \theta^*(\nu)^T \nabla_\theta f_h(\theta^*(\nu)). \qquad (9)$$

From the first order optimality condition for the lower level objective, it holds that

$$\nabla_\theta V_s(\nu, \theta^*(\nu)) = 0, \qquad (10)$$

which is the gradient of lower-level objective with respect to parameter $\theta$ evaluated at the optimal $\theta^*(\nu)$. Now, differentiating again with respect to $\nu$ on both sides of (10), we obtain

$$\nabla^2_{\nu,\theta} V_s(\nu, \theta^*(\nu)) + \nabla_\nu \theta^*(\nu) \nabla^2_\theta V_s(\nu, \theta^*(\nu)) = 0. \quad (11)$$

The above expression would imply that we can write the final expression for the gradient in (9) as

$$\nabla_\nu f_h(\theta^*(\nu)) = -\nabla^2_{v,\theta} V_s(\nu, \theta^*(\nu)) \nabla^2_\theta V_s(\nu, \theta^*(\nu))^{-1}$$

$$(12)$$

$$\nabla_\theta f_h(\theta^*(\nu)).$$

We substitute (12) into (8) to write the final expression for the outer objective in (2):

$$\nabla_\nu G[\nu, \theta^*(\nu)] = \mathbb{E}_{P(\tau;\theta^*(\nu))} \left[ U(\tau) \cdot \sum_{h=0}^{H_u-1} [-\nabla^2_{v,\theta} V_s(\nu, \theta^*(\nu)) \right.$$

$$(13)$$

$$\left. \cdot \nabla^2_\theta V_s(\nu, \theta^*(\nu))^{-1} \nabla_\theta f_h(\theta^*(\nu))] \right]$$

$$+ \nabla_\nu Z(\nu).$$

From the gradient expression in (13), we note that there are three intertwined technical challenges for solving the reward alignment problem efficiently:

(i) requirement of access to $\theta^*(\nu)$

(ii) evaluating Jacobians and Hessians of the lower-level objectives

(iii) unbiasedly sampling trajectories from $\mathbb{P}(\tau;\theta^*(\nu)) = \rho(s_0) \prod_{h=0}^{H_\ell-1} \pi_{\theta^*(\nu)}(a_h|s_h) \mathbb{P}(s'_h \mid \pi_{\theta^*(\nu)}(s_h, a_h))$ dependent on optimal policy $\pi_{\theta^*(\nu)}$

To provide intuition, let us temporarily hypothesize that an oracle provides us access to $\theta^*(\nu)$ for given $\nu$. In this case, we can write a first-order gradient iteration to solve the joint policy learning and alignment problem (2), which is given as Algorithm 2. We note that from the standard analysis of gradient descent, we know that the update in Algorithm 2 would converge to stationary point (for non-convex $G$ and

---

**Algorithm 1** Principal driven Policy Alignment via Bilevel RL (PPA-BRL)

1: **Input**: Reward parametrization $\nu_0$ policy initialization $\theta^0$, upper and lower-level step sizes $\alpha_u > 0, \alpha_\ell > 0$ respectively
2: **for all** $t = 0, 1, 2, ..., T-1$ **do**
3:     **for all** $k = 0, 2, ..., K-1$ **do**
4:         Sample $N$ trajectories $\tau \sim P(\tau; \theta^K(\nu_t))$ and estimate policy gradient $\nabla_\theta V_s(\nu_t, \theta^K(\nu_t))$ from equation (27)
5:         Update the policy gradient parameter as :

$$\theta^{k+1}(\nu_t) = \theta^k(\nu_t) - \alpha_\ell \nabla_\theta V_s(\nu_t, \theta^k(\nu_t))$$

6:     **end for**
7:     Update the reward parameterization in the upper-level from equation (26) as :

$$\nu_{t+1} \leftarrow \nu_t - \alpha_u \widetilde{\nabla}_\nu G(\nu_t, \theta^K(\nu_t))$$

8: **end for**
9: **Output:** $\nu_T, \theta^K(\nu_T)$

---

optimal objective for convex $G$). Once, we have learned the optimal reward parameter $\nu_T$, we can utilize that to obtain optimal policy $\theta^*(\nu_T)$ aligned with the upper-level objectives. But as mentioned earlier, there are impediments to implementing Algorithm 2 in practice. Therefore, we develop a stochastic algorithm without requiring access to $\theta^*(\nu)$ oracle next.

*Remark* 3.1. We have only presented the analytical forms of the first and second-order information required to obtain a numerical solver for problem in (2). However, in practice, these update directions are unavailable due to their dependence on distributions $\mathbb{P}(\tau; \theta^*(\nu))$ and MDP transition model $\mathbb{P}$. Therefore, only sampled estimates of the expressions in (26)-(29) are available.

## 4. Convergence Analysis

In this section, we analyze the convergence behavior of Algorithm 1. Since the outer objective $G(\nu, \theta^*(\nu))$ is non-convex with respect to $\nu$, we consider $\nabla_\nu G(\nu, \theta^*(\nu))$ as our convergence criteria and show its convergence to a first-order stationary point, as well as the convergence of $\theta^K(\nu)$ to $\theta^*(\nu)$. Taken together, these constitute a local KKT point (Boyd & Vandenberghe, 2004)[Ch. 5] Without loss of generality, our convergence analysis is for the minimization (upper and lower level objectives). We proceed then by introducing some technical conditions required for our main results.

*Assumption* 1 (Lipschitz gradient of outer objective). For any $\nu \in \mathbb{R}^n$, the gradient of the outer objective is Lipschitz

continuous w.r.t to second argument with parameter $L_g$, i.e., we may write

$$\|\nabla_\nu G(\nu, \theta^*(\nu)) - \nabla_\nu G(\nu, \theta^K(\nu))\| \leq L_g \|\theta^*(\nu) - \theta^K(\nu)\|. \tag{14}$$

*Assumption 2.* For all $s \in \mathcal{S}$ and $a \in \mathcal{A}$, reward function is bounded as $r(s, a) \leq R$ for all $(s, a)$ and Lipschitz w.r.t to $\nu$, i.e., $|r_{\nu_1}(s, a) - r_{\nu_2}(s, a)| \leq L_r \|\nu_1 - \nu_2\|;$, and bounded $\forall(s, a)$ i.e $r_\nu(s, a) \leq R$.

*Assumption 3.* The policy $\pi_\theta$ is Lipschitz with respect to parameter $\theta$, which implies $\|\pi_{\theta_1}(\cdot|s) - \pi_{\theta_2}(\cdot|s)\| \leq L_\pi \|\theta_1 - \theta_2\|$ for all $\theta_1 \neq \theta_2$. The score function $\nabla_\theta \log \pi_\theta(a|s)$ is bounded $\|\nabla_\theta \log \pi_\theta(a|s)\| \leq B$ and Lipschitz:

$$\|\nabla_\theta \log \pi_{\theta_1}(\cdot|s) - \nabla_\theta \log \pi_{\theta_2}(\cdot|s)\| \leq L_1 \|\theta_1 - \theta_2\| \tag{15}$$

$\forall \theta_1 \neq \theta_2$. Further, the policy parameterization induces score function whose Hessian is Lipschitz:

$$\|\nabla_\theta^2 \log \pi_{\theta_1}(\cdot|s) - \nabla_\theta^2 \log \pi_{\theta_2}(\cdot|s)\| \leq L_2 \|\theta_1 - \theta_2\| \tag{16}$$

$\forall \theta_1 \neq \theta_2$.

*Assumption 4.* $V_s(\nu, \theta)$ satisfies the Polyak-Lojasiewicz (PL) condition with respect to $\theta$ with parameter $\mu$. We denote $\{\lambda(\nabla_\theta^2 V_s(\nu, \theta))_j\}_{j=1}^d$ as the eigenvalues of Hessian matrix $\nabla_\theta^2 V_s(\nu, \theta)$. Although, $V_s(\nu, \theta)$ is non-convex in $\theta$, but follows the restriction on the eigenvalues as $\lambda(\nabla_\theta^2 V_s(\nu, \theta)) \in [-\hat{l}, -\hat{\mu}] \cup [\hat{\mu}, \hat{l}]$.

Next, we introduce a few key technical lemmas regarding the iterates generated by Algorithm 1. We begin by quantifying the distributional drift associated with the transition model under $\theta^K(\nu_t)$ as compared with $\theta^*(\nu_t)$ at the lower level, which results in a transient effect at the outer level.

**Lemma 4.1.** *Under Assumptions 1 - 4, for trajectory $\tau = \{s_h, a_h\}_{h=1}^{H_u}$, it holds that*

$$D_f(P(\tau; \theta^*(\nu)), P(\tau; \theta^K(\nu))) \leq \frac{H_u L_2}{2} \|\theta^*(\nu) - \theta^K(\nu)\|, \tag{17}$$

*where $D_f$ is the f-divergence between distributions and $L_\pi$ is the Lipschitz parameter (cf. Assum. 3).*

The proof of Lemma 4.1 in provided in Appendix 11.1. Next, we establish some error bound conditions on key second-order terms that appear in equations (26)-(29) when we substitute $\theta^*(\nu_t)$ by $\theta^K(\nu_t)$.

**Lemma 4.2** (Value function related upper bounds). *Under Assumptions 1 - 4, it holds that*

(i) *The second order Jacobian term $\nabla_{\nu,\theta}^2 V_s(\nu_t, \theta^K(\nu_t))$ is bounded as $\|\nabla_{\nu,\theta}^2 V_s(\nu_t, \theta^K(\nu_t))\| \leq H_\ell^2 L_r B$, where*

$H_\ell$ *is the horizon length for the lower level [cf. (2)], $L_r$ is the reward Lipschitz parameter [cf. Assumption 2], and $B$ is the score function bound [cf. Assumption 3].*

(ii) *The Hessian of the value function is Lipschitz with parameter as*

$$\|\nabla_\theta^2 V_s(\nu, \theta^*(\nu)) - \nabla_\theta^2 V_s(\nu, \theta^K(\nu))\| \tag{18}$$
$$\leq L' \|\theta^*(\nu) - \theta^K(\nu)\|,$$

*where, $L' = L_{f_1} \chi_1 \frac{H_\ell}{2} L_2 + L_{f_1}$ and $L_{f_1} = L_2 H_\ell^2 R$. Here, $H_\ell$ is the horizon length at the lower level, $\chi_1$ is a constant defined in (64), $R$ is the maximum reward (cf. Assumption 2), and $L_2$ is the Lipschitz parameter of the Hessian of the policy defined in Assumption 3.*

(iii) *The second order mixed jacobian term $\nabla_{\nu,\theta}^2 V_s(\nu, \theta^K(\nu))$ is Lipschitz continuous w.r.t $\theta$ i.e*

$$\|\nabla_{\nu,\theta}^2 V_s(\nu, \theta^*(\nu_t)) - \nabla_{\nu,\theta}^2 V_s(\nu, \theta^K(\nu))\| \tag{19}$$
$$\leq L'' \|\theta^*(\nu) - \theta^K(\nu)\|$$

*where, $L'' = L_{f_3} \chi_2 \frac{H_\ell}{2} L_2 + L_{f_3}$ and $L_{f_3} = L_r L_1 H_\ell^2$. Here, $\chi_2$ is a constant defined in equation (72), and other constants are as defined in statement (ii).*

The proof of Lemma 4.2 is in Appendix 11.2. To prove this result, we start by considering the value function expression, and evaluating it's gradient, Hessian, and Jacobians. After expanding each of them, we separate the reward and policy-related terms and then utilize the aforementioned assumptions to upper bound them, respectively.

**Lemma 4.3.** *Let us define the update direction associated with the gradient in (13) as*

$$\phi_1(\tau) := U(\tau) \cdot \sum_{t=0}^{H_u} [-\nabla_{v,\theta}^2 V_s(\nu, \theta^*(\nu)) \tag{20}$$
$$\cdot \nabla_\theta^2 V_s(\nu, \theta^*(\nu))^{-1} \nabla_\theta f_h(\theta^*(\nu))],$$

$$and \ \phi_2(\tau) := U(\tau) \cdot \sum_{t=0}^{H_u} [-\nabla_{v,\theta}^2 V_s(\nu, \theta^K(\nu)) \tag{21}$$
$$\cdot \nabla_\theta^2 V_s(\nu, \theta^K(\nu))^{-1} \nabla_\theta f_h(\theta^K(\nu))].$$

*Under Assumptions 1 - 4, it holds that*

$$\|\mathbb{E}_{\tau \sim P(\tau; \theta^K(\nu))}[\phi_1(\tau) - \phi_2(\tau)]\| \leq H_u^2 \tilde{u} \gamma_1 \|\theta^*(\nu) - \theta^K(\nu)\|, \tag{22}$$

*where $\gamma_1 := \kappa L_1 + \frac{L_{\nu,\theta} L'}{l_\pi^2} + \frac{L''}{l_\pi}$. Here, $\tilde{u}$ is the upper bound of utility $U(\tau)$ defined in (5), $l_\pi, L_1$ are policy-related Lipschitz parameters (cf. Assumption 3), $L'$ and $L''$ as defined in Lemma 4.2, and $\kappa$ mixed condition number defined in equation (92) and $L_{\nu,\theta}$ upper-bound of the norm of second order mixed jacobian term, defined in equation (90)*

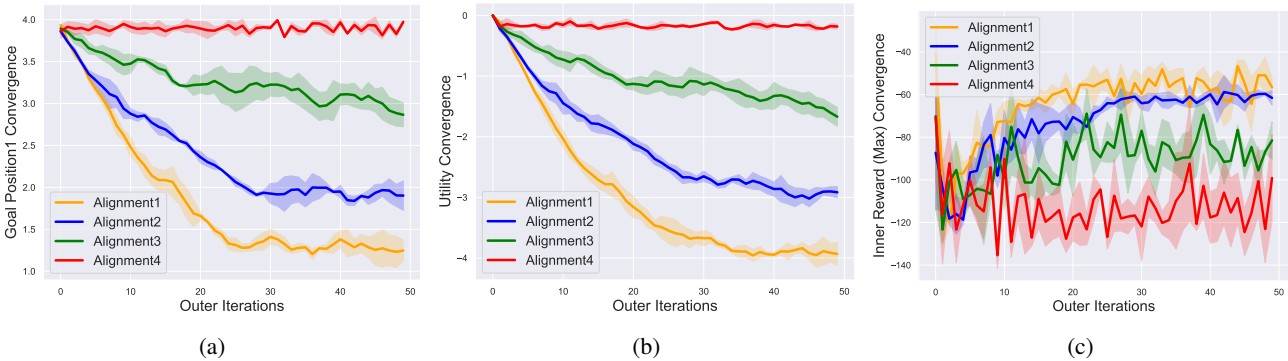

*Figure 1.* **(a)** describes the convergence of the upper-level optimization variable with the designer's alignment levels. Alignments 1, 2, 3, and 4 used $\lambda = 1, 10, 15, 30$, respectively. It is evident that our Algorithm aligns efficiently with the total outer level utility i.e considering both the aspects of outer and inner utilities for the designer. Figure 1(b) shows that when weightage on the cost-utility term in the outer objective is less, the cost naturally increases with outer iterations (shown as a negative reward). Figure 1(c) shows that for alignment configurations with low weights on the cost-utility part of the outer objective, inner rewards are achieved is more and vice-versa otherwise. Overall it demonstrates that our algorithm efficiently aligns to the broader utilities

The proof of Lemma 4.3 is in Appendix 12.6.

**Lemma 4.4.** *Under Assumptions 1-(4), the lower level iterates of Algorithm 1 satisfies*

$$\|\theta^K(\nu_t) - \theta^*(\nu)\|^2 \le \frac{\eta^K L_6}{\mu} Z, \qquad (23)$$

*where,* $Z := \max_\nu \|\theta^0 - \theta^*(\nu)\|^2$, $\eta := 1 - \alpha_3$, $\alpha_3 = \alpha_\ell(1 - \frac{\alpha_\ell L_6}{2})\frac{\mu}{2}$, $L_6 = L_5 \frac{H L_2}{2} + L_5$, $L_5 = H_l^2 R L_1$, $\mu$ *is the PL constant, and K denotes the number of lower-level iterations, and policy gradient step-size satisfies* $\alpha_\ell < 2/\min(H_\ell L_2, \mu)$, *with $\mu$ as the PL constant (Assumption 4).*

The proof of Lemma 4.4 is provided in Appendix 12.7. The proof relies on the assumption that the Value function satisfies a Polyak Łojasiewicz (PL) condition under appropriate policy parametrization.

*Theorem* 1. Under Assumptions 1-(4), for the proposed Algorithm 1, it holds that

$$\frac{1}{T}\sum_{t=1}^{T}\|\nabla_\nu G(\nu_t, \theta^*(\nu_t))\|^2 \le \frac{G_0 - G^*}{\delta_1 T} + \frac{\eta \delta_2 L_6 Z}{T \delta_1 \mu (1 - \eta)}$$

$$(24)$$

where $G_0 := G(\nu_0, \theta^*(\nu_0))$ and $G^*$ denotes the global optimum of the outer objective, $\delta_1 = \alpha_u \left(1 - \frac{1}{2c_1} - L_g \alpha_u\right)$ and $\delta_2 = \alpha_u \left(\frac{c_1}{2} + L_g \alpha_u\right)$, $c_1$ is a positive constant defined in eqn. (33), and the step-size range of satisfies $\alpha_u < 1/L_g$, with $L_g$ as in Assumption 1 and $\alpha_\ell$ as stated in Lemma 4.4.

In Theorem 1, we note that we achieve a final rate of $\mathcal{O}(1/T)$, which matches with bilevel optimization for non-convex outer objective (Ghadimi & Wang, 2018a) when the lower-level satisfies PL condition.

## 5. Experimental Evaluations

To demonstrate our algorithm's performance, we use a simple grid-world environment where the agent's objective is to reach a goal at position $(G_x, G_y)$. The agent starts at $(0,0)$ and receives a reward $r_\nu$ based on its speed in reaching the goal. The goal's position affects an outer utility/cost $R_{util}$, which incurs a cost of $-\sqrt{(T_x - G_x)^2 + (T_y - G_y)^2}$ as it moves away from the target position $(T_x, T_y)$. We formulate this as a bilevel optimization problem with an overall cost $R_{outer} = R_{inner} + \lambda R_{util}$, where $\lambda$ is a scalar weighting term. Smaller $\lambda$ values prioritize inner utility, causing the goal to move closer to the agent's start position. Increasing $\lambda$ gives more weight to outer utility, bringing the goal closer to the target position. Fig.1a shows goal position changes under different alignments, while Fig.1b and Fig.1c depict the convergence of outer and inner utility to the alignments. Alignments 1, 2, 3, and 4 correspond to $\lambda = 1, 10, 15, 30$ respectively. Lower $\lambda$ values lead to a goal closer to the agent and higher inner utility, while higher $\lambda$ values keep the goal near the target with higher outer utility. The ablation study validates the algorithm's alignment performance.

## 6. Conclusions and Future Work

Potentially misaligned agents pose a severe risk to society; thus making AI alignment at the forefront of research of the current times (Liu et al., 2022). However, an efficient solution to such an alignment problem depends upon two major factors 1) precise Evaluation of the agent's policy and 2) subsequent reward design, and failure to adhere to any of these can cause potentially misaligned agents or agents with compromised performance. To deal with the above entanglement, we formulate the alignment problem in a bilevel optimization framework and characterized a precise

evaluation by approximately estimating the gradient of the optimal policy w.r.t to the reward parametrization, which is crucial for alignment. Finally, through our design and analysis of the algorithm, we show convergence guarantees to the broader goals with a rate $\mathcal{O}(1/T)$, which is a first step towards provable policy alignment research.

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

# Appendix

## 7. Notations

We collectively describe the notations used in this work in Table 1.

Table 1:

| Notations | Details |
| --- | --- |
| $\mathcal{S}, \mathcal{A}$ | State space, action space |
| $(s, a)$ | State-action pair |
| $\mathbb{P}(s'\|s, a)$ | Transition kernel |
| $r_\nu(s, a)$ | Reward function parameterized by designer's parameters $\nu \in \mathbb{R}^n$ |
| $\pi_{\theta(\nu)}(\cdot\|s)$ | Policy parameterized by $\theta(\nu) \in \mathbb{R}^d$ for design parameters $\nu$ |
| $V_s(\nu, \theta(\nu))$ | Inner objective - Value function for state $s$ at outer parameter $\nu$ and policy parameter $\theta(\nu)$ |
| $G(\nu, \theta^*(\nu))$ | Outer objective |

## 8. Related Work

**Bilevel Optimization.** Multi-stage optimization has a long-history in optimization and operations research, both for deterministic (Bertsimas & Caramanis, 2010) and stochastic objectives (Pflug & Pichler, 2014). In general, these problems exhibit NP-hardness if the objectives are general non-convex functions. Therefore, much work in this area restricts focus to classes of problems, such as affine (Bertsimas & Caramanis, 2010; Bertsimas et al., 2010). From both the mathematical optimization community (Bracken & McGill, 1973) and algorithmic game theory (Von Stackelberg, 2010), extensive interest has been given to specifically two-stage problems. A non-exhaustive list contains the following works (Ghadimi & Wang, 2018b; Yang et al., 2021; Khanduri et al., 2021; Li et al., 2022; Ji & Liang, 2022; Huang et al., 2022). Predominately, one these works do not consider that either stage is a MDP, and therefore while some of the algorithmic strategies are potentially generalizable to the RL agent alignment problem, they are not directly comparable to the problem studied here.

**Algorithms for Stackleberg Games.** Algorithmic methods for Sackleberg games are a distinct line of inquiry that develops methods to reach the Stackleberg-Nash equilibrium of a game, which have received significant attention in recent years. Beginning with the static Stackleberg game setting, conditions for gradient play to achieve local equilibria have been established in (Fiez et al., 2019). Follow on work studied conditions for gradient play to converge without convexity, but does not allow for MDP/trajectory dependence of objective functions at either stage (Maheshwari et al., 2023). On the other hand, the access to information structures have gradually been relaxed to allow bandit feedback (Bai et al., 2021) and linear MDPs (Zhong et al., 2021). Value iteration schemes have been proposed to achieve Stackleberg-Nash equilibria as well (Goktas et al., 2022), although it is unclear how to generalize them to handle general policy parameterization in a scalable manner. Most similar to our work are those that develop implicit function-theorem based gradient play (Fiez et al., 2020; Vu et al.); however, there are no performance certificates for these approaches.

**Mechanism Design.** In this line of research, one studies the interrelationship between the incentives of an individual economic actor and their macro-level behavior at the level of a social welfare objective. This literature can be traced back to (Myerson, 1989; Hurwicz, 2003; Maskin, 2008), and typically poses the problem as one that does not involve sequential interactions. More recently, efforts to cast the evolution of the outer-stage which quantifies social welfare or ethical considerations as a sequential process, i.e., an MDP, have been considered (Tang, 2017; Hu et al., 2018). In these works, agents' behavior is treated as fixed and determining of the state transition dynamics, which gives rise to a distinct subclass of policy optimization problems (Lyu et al., 2022a;b).

**Principal-Agent Problem.** Contract theory also formalizes the notion of an agent's interaction with a rule setter (Green & Stokey, 1983; Eisenhardt, 1989; Stiglitz, 1989). The agent is the individual who may choose to enroll in a contract, and the principal is the creator of the rules or incentive structure that determines the incentive structure for the agent's decision. Formalizing the question of how to design a contract and agent behavior through MDP machinery and Stackleberg games

---

**Algorithm 2** Policy Alignment with Reward Design (conceptual)

---

1: **Input**: Reward parametrization $\nu_0$, optimal policy parameter $\theta^*(\nu_0)$ (oracle), upper-level step size $\alpha > 0$
2: **for all** $t = 0, 1, 2, ..., T - 1$ **do**
3:     Update the parameter $\nu_t$ as

$$\nu_{t+1} = \nu_t - \alpha \nabla_\nu G[\nu_t, \theta^*(\nu_t)] \tag{25}$$

    where $\nabla_\nu G[\nu, \theta^*(\nu_t)]$ is defined in (13).
4: **end for**
5: **Output**: $\nu_T$

---

has been the subject of recent efforts, especially (Zhu et al., 2022; Chen et al., 2023b).

**RL with Preferences.** Revealed preferences is a concept in economics that states that humans in many cases do not know what they prefer until they are exposed to examples (Caplin & Dean, 2015). This idea gained traction as a it provided a substantive basis for querying humans and elucidating feedback on decision-making problems with metrics whose quantification is not obvious, such as trustworthiness or fairness. Efforts to incorporate pairwise comparison into reinforcement learning, especially as a mechanism to incorporate preference information, has been heavily studied in recent years – see (Wirth et al., 2017) for a survey. A non-exhaustive list of works along these lines is (Isbell et al., 2001; Fürnkranz et al., 2012; Roth et al., 2016; Hill et al., 2021; Wirth et al., 2017; Zhu et al., 2023; Xu et al., 2020; Saha et al., 2023).

**Inverse RL and Behavioral Cloning.** Implicitly contained in the RL with preferences framework is the assumption that humans should design a reward function to drive an agent towards the correct behavior through its policy optimization process. This question of reward design has also been studied through an alternative lens in which one provides demonstrations or trajectories, and seeks to fit a reward function to represent this information succinctly. This class of problems, broadly referred to as inverse RL (Ng et al., 2000), has a long history in experimentally driven RL research – see (Arora & Doshi, 2021) for a survey. Rather than design a reward function as an intermediary between demonstrations and policy optimization, behavioral cloning directly seeks to mimic the behavior of demonstration information (Torabi et al., 2018; Wen et al., 2020). In doing so, it is able to sidestep some of the questions of whether a reward function can be well-posed for a given collection of trajectories.

## 9. Upper & Lower Level Gradient Computations

**Upper-level Gradient Estimation** To estimate the gradient of the upper-level objective in equation (8), we require information about $\theta^*(\nu)$ (issue (i)) which is not available in general unless the inner-level objective has a closed-form solution. Hence, we approximate $\theta^*(\nu_t)$ with $\theta^K(\nu_t)$ i.e., running K-step policy gradient steps at the lower-level as

$$\widetilde{\nabla}_\nu G(\nu_t, \theta^K(\nu_t)) = \mathbb{E}_{P(\tau;\theta^K(\nu_t))} \left[ U(\tau) \cdot \sum_{t=0}^{H_u - 1} [\widetilde{M}^K(\nu_t) \nabla_\theta f_h(\theta^K(\nu_t))] \right] + \nabla_\nu Z(\nu). \tag{26}$$

where $\widetilde{M}^K(\nu_t) = -\nabla_{v,\theta}^2 V_s(\nu_t, \theta^K(\nu_t)) \nabla_\theta^2 V_s(\nu_t, \theta^K(\nu_t))^{-1}$. This approximate evaluation of (13) with a current policy estimate $\theta^K(\nu_t)$ in lieu of $\theta^*(\nu_t)$ constitutes our effort to address issue (iii).

**Lower-level objective gradient, Jacobian, and Hessian estimation:** Next, we derive the exact gradients for the inner-level objective and, subsequently the Hessian and mixed hessian terms for our algorithmic description (issue (ii)). Let us first write down the gradient of lower-level objectives using the Policy Gradient Theorem (Williams, 1992; Sutton et al., 1999) as

$$\nabla_\theta V_s(\nu_t, \theta^K(\nu_t)) = \mathbb{E}_{P(\tau;\theta^K(\nu_t))} \left[ \sum_{h=0}^{H_\ell - 1} \gamma^h r_{\nu_t}(s_h, a_h) \left( \sum_{j=0}^{H_\ell - 1} \nabla_\theta \log \pi_{\theta^K(\nu_t)}(a_j | s_j) \right) \right], \tag{27}$$

where $H_\ell$ is the horizon or the episode length. Similarly, the Hessian of the inner objective is:

$$\nabla_\theta^2 V_s(\nu_t, \theta^K(\nu_t)) = \mathbb{E}_{P(\tau;\theta^K(\nu_t))} \left[ \sum_{h=0}^{H_\ell-1} \gamma^h r_{\nu_t}(s_h, a_h) \left( \sum_{j=0}^{H_\ell-1} \nabla_\theta^2 \log \pi_{\theta^K(\nu_t)}(a_j|s_j) \right) \right], \tag{28}$$

Finally, we can write the mixed second-order Jacobian matrix as

$$\nabla_{\nu,\theta}^2 V_s(\nu_t, \theta^K(\nu_t)) = \mathbb{E}_{P(\tau;\theta^K(\nu_t))} \left[ \sum_{h=0}^{H_\ell-1} \gamma^h \nabla_\nu r_{\nu_t}(s_h, a_h) \left( \sum_{j=0}^{H_\ell-1} [\nabla_\theta \log \pi_{\theta^K(\nu_t)}(a_j|s_j)]^T \right) \right]. \tag{29}$$

Now, utilizing the expressions in (26)-(29), we summarize the proposed steps in Algorithm 1. We note that we denote the number of inner-level iterates by $K$ for better exposition. We will choose it as a function of outer iterations $t$ in the convergence analysis next. Without loss of generality, we have written the algorithm updates as if we are minimizing the upper and lower level objectives. Before shifting to analyzing the convergence behavior of Algorithm 1, we close with a remark.

## 10. Proof of Theorem 1

*Proof.* We begin by the smoothness assumption in the upper-level objective (cf. Assumption 14), which implies that

$$G[\nu_{t+1}, \theta^*(\nu_{t+1})] \leq G(\nu_t, \theta^*(\nu_t)) + \langle \nabla_\nu G(\nu_t, \theta^*(\nu_t)), \nu_{t+1} - \nu_t \rangle + \frac{L_g}{2} \|\nu_{t+1} - \nu_t\|^2. \tag{30}$$

From the update of outer parameter $\nu_{t+1}$ (cf. (26)), we holds that

$$\begin{aligned} G(\nu_{t+1}, \theta^*(\nu_{t+1})) \leq &G(\nu_t, \theta^*(\nu_t)) + \langle \nabla_\nu G(\nu_t, \theta^*(\nu_t)), -\alpha_u \widetilde{\nabla}_\nu G(\nu_t, \theta^K(\nu_t)) \rangle \\ &+ \frac{L_g \alpha_u^2}{2} \|\widetilde{\nabla}_\nu G(\nu_t, \theta^K(\nu_t))\|^2. \end{aligned} \tag{31}$$

We add subtract the original gradient $\nabla_\nu G(\nu_t, \theta^*(\nu_t))$ [cf. (13)] in (31) as follows

$$\begin{aligned} G(\nu_{t+1}, \theta^*(\nu_{t+1})) \leq &G(\nu_t, \theta^*(\nu_t)) + \langle \nabla_\nu G(\nu_t, \theta^*(\nu_t)), -\alpha_u \nabla_\nu G(\nu_t, \theta^*(\nu_t)) \rangle \\ &+ \alpha_u \langle \nabla_\nu G(\nu_t, \theta^*(\nu_t)), \nabla_\nu G(\nu_t, \theta^*(\nu_t)) - \widetilde{\nabla}_\nu G(\nu_t, \theta^K(\nu_t)) \rangle \\ &+ \frac{L_g \alpha_u^2}{2} \|\nabla_\nu G(\nu_t, \theta^*(\nu_t)) + \widetilde{\nabla}_\nu G(\nu_t, \theta^K(\nu_t)) - \nabla_\nu G(\nu_t, \theta^*(\nu_t))\|^2 \\ = &G(\nu_t, \theta^*(\nu_t)) - \alpha_u \|\nabla_\nu G(\nu_t, \theta^*(\nu_t))\|^2 \\ &+ \alpha_u \langle \nabla_\nu G(\nu_t, \theta^*(\nu_t)), \nabla_\nu G(\nu_t, \theta^*(\nu_t)) - \widetilde{\nabla}_\nu G(\nu_t, \theta^K(\nu_t)) \rangle \\ &+ \frac{L_g \alpha_u^2}{2} \|\nabla_\nu G(\nu_t, \theta^*(\nu_t)) + \widetilde{\nabla}_\nu G(\nu_t, \theta^K(\nu_t)) - \nabla_\nu G(\nu_t, \theta^*(\nu_t))\|^2. \end{aligned} \tag{32}\ (33)$$

Using Peter-Paul inequality for the third term on the right hand side of (33), we get

$$\begin{aligned} G(\nu_{t+1}, \theta^*(\nu_{t+1})) \leq &G(\nu_t, \theta^*(\nu_t)) - \alpha_u \|\nabla_\nu G(\nu_t, \theta^*(\nu_t))\|^2 + \frac{\alpha_u}{2c_1} \|\nabla_\nu G(\nu_t, \theta^*(\nu_t))\|^2 \\ &+ \frac{\alpha_u c_1}{2} \|\nabla_\nu G(\nu_t, \theta^*(\nu_t)) - \widetilde{\nabla}_\nu G(\nu_t, \theta^K(\nu_t))\|^2 \\ &+ \frac{L_g \alpha_u^2}{2} \|\nabla_\nu G(\nu_t, \theta^*(\nu_t)) + \widetilde{\nabla}_\nu G(\nu_t, \theta^K(\nu_t)) - \nabla_\nu G(\nu_t, \theta^*(\nu_t))\|^2. \end{aligned} \tag{34}$$

where $c_1 \geq 0$. Next, after grouping the terms, we get

$$
\begin{aligned}
G(\nu_{t+1}, \theta^*(\nu_{t+1})) \leq & G(\nu_t, \theta^*(\nu_t)) - \alpha_u \left( 1 - \frac{1}{2c_1} \right) \|\nabla_\nu G(\nu_t, \theta^*(\nu_t))\|^2 \\
& + \frac{\alpha_u c_1}{2} \|\nabla_\nu G(\nu_t, \theta^*(\nu_t)) - \widetilde{\nabla}_\nu G(\nu_t, \theta^K(\nu_t))\|^2 \\
& + L_g \alpha_u^2 \|\nabla_\nu G(\nu_t, \theta^*(\nu_t))\|^2 + L_g \alpha_u^2 \|\widetilde{\nabla}_\nu G(\nu_t, \theta^K(\nu_t)) - \nabla_\nu G(\nu_t, \theta^*(\nu_t))\|^2 \\
= & G(\nu_t, \theta^*(\nu_t)) - \alpha_u \left( 1 - \frac{1}{2c_1} - L_g \alpha_u \right) \|\nabla_\nu G(\nu_t, \theta^*(\nu_t))\|^2 \\
& + \alpha_u \left( \frac{c_1}{2} + L_g \alpha_u \right) \|\nabla_\nu G(\nu_t, \theta^*(\nu_t)) - \widetilde{\nabla}_\nu G(\nu_t, \theta^K(\nu_t))\|^2,
\end{aligned}
\tag{35}
$$

where we use the inequality $\|a + b\|^2 \leq 2\|a\|^2 + 2\|b\|^2$, followed by algebraic operations to get the final expression of equation (35). Next, we analyze the term $\|\nabla_\nu G(\nu_t, \theta^*(\nu_t)) - \widetilde{\nabla}_\nu G(\nu_t, \theta^K(\nu_t))\|^2$ from equation (35). Let us start by considering the explicit expressions of $\nabla_\nu G(\nu, \theta^*(\nu))$ and $\widetilde{\nabla}_\nu G(\nu_t, \theta^K(\nu_t))$ in equations (13) and (26) as

$$
\nabla_\nu G(\nu, \theta^*(\nu)) - \widetilde{\nabla}_\nu G(\nu, \theta^K(\nu)) = \mathbb{E}_{\tau \sim P(\tau; \theta^*(\nu))}[\phi_1(\tau)] - \mathbb{E}_{\tau \sim P(\tau; \theta^K(\nu))}[\phi_2(\tau)],
\tag{36}
$$

where we define

$$
\phi_1(\tau) = U(\tau) \cdot \sum_{t=0}^{H_u} [-\nabla_{v,\theta}^2 V_s(\nu, \theta^*(\nu)) \nabla_\theta^2 V_s(\nu, \theta^*(\nu))^{-1} \nabla_\theta f_h(\theta^*(\nu))],
\tag{37}
$$

$$
\text{and} \quad \phi_2(\tau) = U(\tau) \cdot \sum_{t=0}^{H_u} [-\nabla_{v,\theta}^2 V_s(\nu, \theta^K(\nu)) \nabla_\theta^2 V_s(\nu, \theta^K(\nu))^{-1} \nabla_\theta f_h(\theta^K(\nu))].
\tag{38}
$$

Now, we expand the terms in (36) by adding subtracting the term $\mathbb{E}_{\tau \sim P(\tau; \theta^K(\nu))}[\phi_1(\tau)]$ in the right hand side as follows

$$
\begin{aligned}
\nabla_\nu G(\nu, \theta^*(\nu)) - \widetilde{\nabla}_\nu G(\nu, \theta^K(\nu)) = & \mathbb{E}_{\tau \sim P(\tau; \theta^*(\nu))}[\phi_1(\tau)] - \mathbb{E}_{\tau \sim P(\tau; \theta^K(\nu))}[\phi_1(\tau)] \\
& + \mathbb{E}_{\tau \sim P(\tau; \theta^K(\nu))}[\phi_1(\tau)] - \mathbb{E}_{\tau \sim P(\tau; \theta^K(\nu))}[\phi_2(\tau)] \\
= & \mathbb{E}_{\tau \sim P(\tau; \theta^*(\nu))}[\phi_1(\tau)] - \mathbb{E}_{\tau \sim P(\tau; \theta^K(\nu))}[\phi_1(\tau)] \\
& + \mathbb{E}_{\tau \sim P(\tau; \theta^K(\nu))}[\phi_1(\tau) - \phi_2(\tau)].
\end{aligned}
\tag{39, 40}
$$

We note that the first term on the right-hand side of (40) $\mathbb{E}_{\tau \sim P(\tau; \theta^*(\nu))}[\phi_1(\tau)] - \mathbb{E}_{\tau \sim P(\tau; \theta^K(\nu))}[\phi_1(\tau)] \leq \sup_\phi \mathbb{E}_{\tau \sim P(\tau; \theta^*(\nu))}[\phi_1(\tau)] - \mathbb{E}_{\tau \sim P(\tau; \theta^K(\nu))}[\phi_1(\tau)]$. This boils down to the standard definition of Integral Probability Metric(IPM) which, under suitable assumption on the function $\phi$ can be upper-bounded by Wasserstein and Total Variation distance. Specifically, we show that function $\phi$ is Lipschitz with some constant. The second term on the right-hand side of (40) is the expected difference between the two functions. Hence, we can write (40) as

$$
\begin{aligned}
\nabla_\nu G(\nu, \theta^*(\nu)) - \widetilde{\nabla}_\nu G(\nu, \theta^K(\nu)) = & D_f(P(\tau; \theta^*(\nu)), P(\tau; \theta^K(\nu))) \\
& + \mathbb{E}_{\tau \sim P(\tau; \theta^K(\nu))}[\phi_1(\tau) - \phi_2(\tau)],
\end{aligned}
\tag{41}
$$

where $D_f$ denotes the f-divergence between two distributions. Taking the norm on both sides and from the statements of Lemma 4.1 and Lemma 4.3, we can write

$$
\|\nabla_\nu G(\nu, \theta^*(\nu)) - \widetilde{\nabla}_\nu G(\nu, \theta^K(\nu))\|^2 \leq \left( \frac{H_u^2 L_2^2}{2} + 2 H_u^4 \tilde{u}^2 \gamma_1^2 \right) \|\theta^*(\nu) - \theta^K(\nu)\|^2.
\tag{42}
$$

Utilizing this bound in (35), we get

$$
\begin{aligned}
G(\nu_{t+1}, & \theta^*(\nu_{t+1})) - G(\nu_t, \theta^*(\nu_t)) \\
& \leq -\alpha_u \left( 1 - \frac{1}{2c_1} - L_g \alpha_u \right) \|\nabla_\nu G(\nu_t, \theta^*(\nu_t))\|^2 \\
& \quad + \alpha_u \left( \frac{c_1}{2} + L_g \alpha_u \right) \left( \frac{H_u^2 L_2^2}{2} + 2 H_u^4 \tilde{u}^2 \gamma_1^2 \right) \|\theta^*(\nu) - \theta^K(\nu)\|^2,
\end{aligned}
\tag{43}
$$

where we write the final expression for the convergence analysis from equation (35) and for simplicity of notations, let's assume $\delta_1 = \alpha_u \left(1 - \frac{1}{2c_1} - L_g \alpha_u\right)$ and $\delta_2 = \alpha_u \left(\frac{c_1}{2} + L_g \alpha_u\right)\left(\frac{H_u^2 L_2^2}{2} + 2H_u^4 \tilde{u}^2 \gamma_1^2\right)$, which leads to the simplified version of the equation (43)

$$G(\nu_{t+1}, \theta^*(\nu_{t+1})) - G(\nu_t, \theta^*(\nu_t)) \leq -\delta_1 \|\nabla_\nu G(\nu_t, \theta^*(\nu_t))\|^2 + \delta_2 \|\theta^*(\nu_t) - \theta^K(\nu_t)\|^2. \tag{44}$$

From the statement of Lemma 4.4, we can upper bound the above expression as

$$G(\nu_{t+1}, \theta^*(\nu_{t+1})) - G(\nu_t, \theta^*(\nu_t)) \leq -\delta_1 \|\nabla_\nu G(\nu_t, \theta^*(\nu_t))\|^2 + \delta_2 \frac{\eta^K L_6}{\mu} Z, \tag{45}$$

where we know $\eta \in (0, 1)$. Next, we select $K = t + 1$ to obtain

$$G(\nu_{t+1}, \theta^*(\nu_{t+1})) - G(\nu_t, \theta^*(\nu_t)) \leq -\delta_1 \|\nabla_\nu G(\nu_t, \theta^*(\nu_t))\|^2 + \delta_2 \frac{\eta^{t+1} L_6}{\mu} Z. \tag{46}$$

Taking the summation over $t = 0$ to $T - 1$ on both sides, we get

$$G(\nu_T, \theta^*(\nu_T)) - G(\nu_0, \theta^*(\nu_0)) \leq -\delta_1 \sum_{t=0}^{T-1} \|\nabla_\nu G(\nu_t, \theta^*(\nu_t))\|^2 + \frac{\delta_2 L_6 Z}{\mu} \sum_{t=0}^{T-1} \eta^{t+1} \tag{47}$$

After rearranging the terms, we get

$$\sum_{t=0}^{T-1} \|\nabla_\nu G(\nu_t, \theta^*(\nu_t))\|^2 \leq \frac{G(\nu_0, \theta^*(\nu_0)) - G(\nu_T, \theta^*(\nu_T))}{\delta_1} + \frac{\eta \delta_2 L_6 Z}{\delta_1 \mu} \sum_{t=0}^{T-1} \eta^t$$
$$\leq \frac{G(\nu_0, \theta^*(\nu_0)) - G(\nu_T, \theta^*(\nu_T))}{\delta_1} + \frac{\eta \delta_2 L_6 Z}{\delta_1 \mu (1 - \eta)}. \tag{48}$$

Let us denote $G_0 := G(\nu_0, \theta^*(\nu_0))$ and upper bound $-G(\nu_T, \theta^*(\nu_T)) \leq -G^*$ where $G^*$ denotes the global optimal value of the outer objective. After dividing both sides in (48) by $T$, we get

$$\frac{1}{T} \sum_{t=0}^{T-1} \|\nabla_\nu G(\nu_t, \theta^*(\nu_t))\|^2 \leq \frac{G_0 - G^*}{\delta_1 T} + \frac{\eta \delta_2 L_6 Z}{T \delta_1 \mu (1 - \eta)}. \tag{49}$$

$\square$

## 11. Proof of All Lemmas 4.1-4.4

### 11.1. Proof of Lemma 4.1

*Proof.* The probability distribution of the trajectory $\tau = \{s_h, a_h\}_{h=1}^H$ is given by

$$P(\tau; \theta^*(\nu)) = \rho(s_0) \prod_{h=1}^H \pi_{\theta^*(\nu)}(a_h|s_h) \mathbb{P}(s_{h+1}|s_h, a_h). \tag{50}$$

Similarly, we can derive an equivalent expression for the probability of trajectory induced by the policy $\pi_{\theta^K(\nu)}$ by replacing $\theta^*(\nu)$ with $\theta^K(\nu)$. Here, $\mathbb{P}(s_{h+1}|s_h, a_h)$ is the transition probability which remains the same for both and $\rho(s_0)$ is the initial state distribution. Next, the f-divergence between the two distributions $D_f(P(\tau; \theta^*(\nu)), P(\tau; \theta^K(\nu)))$ can be written as

$$D_f(P(\tau; \theta^*(\nu)), P(\tau; \theta^K(\nu))) \leq \underbrace{D_f(P(\tau; \theta^*(\nu)), P(\tau; \beta))}_{I} + \underbrace{D_f(P(\tau; \beta), P(\tau; \theta^K(\nu)))}_{II}, \tag{51}$$

which holds by triangle inequality (of f-divergences). $P(\tau; \beta)$ represents the trajectory probability induced by another hybrid policy $\pi_\beta(\cdot|s)$ which executes the action based on the policy $\pi_{\theta^K(\nu)}(\cdot|s)$ for the first time-step and then follows the

policy $\pi_{\theta^*(\nu)}(\cdot|s)$ for subsequent timesteps. Now, we focus on term I in (51), we get

$$
\begin{aligned}
D_f(P(\tau;\theta^*(\nu)), P(\tau;\beta)) & \\
&= \sum_\tau P(\tau;\beta) f\left(\frac{P(\tau;\theta^*(\nu))}{P(\tau;\beta)}\right) \\
&= \sum_\tau P(\tau;\beta)) f\left(\frac{\rho(s_0)\pi_{\theta^*(\nu)}(a_1|s_0)\mathbb{P}(s_2|s_1,a_1)\prod_{h=2}^H \pi_{\theta^*(\nu)}(a_h|s_h)\mathbb{P}(s_{h+1}|s_h,a_h)}{\rho(s_0)\pi_{\theta^K(\nu)}(a_1|s_0)\mathbb{P}(s_2|s_1,a_1)\prod_{h=2}^H \pi_{\theta^*(\nu)}(a_h|s_h)\mathbb{P}(s_{h+1}|s_h,a_h)}\right) \\
&= \sum_\tau P(\tau;\beta)) f\left(\frac{\pi_{\theta^*(\nu)}(a_1|s_0)}{\pi_{\theta^K(\nu)}(a_1|s_0)}\right),
\end{aligned}
\tag{52}
$$

where first we expand upon the definition of the trajectory distribution induced by both policies and get the final expression of the equation (52). By expanding the term $P(\tau;\beta)$ in (52), we obtain

$$
\begin{aligned}
D_f(P(\tau;\theta^*(\nu)), P(\tau;\beta)) &= \sum_{s_0} \rho(s_0) \sum_{a_1} \pi_{\theta^K(\nu)}(a_1|s_0) f\left(\frac{\pi_{\theta^*(\nu)}(a_1|s_1)}{\pi_{\theta^K(\nu)}(a_1|s_0)}\right) \sum_{s_1} \mathbb{P}(s_1|s_0,a_1) \cdots \\
&= \sum_s \rho(s) \sum_a \pi_{\theta^K(\nu)}(a|s) f\left(\frac{\pi_{\theta^*(\nu)}(a|s)}{\pi_{\theta^K(\nu)}(a|s)}\right) \\
&= \mathbb{E}_{\rho(s)} \sum_a \pi_{\theta^K(\nu)}(a|s) f\left(\frac{\pi_{\theta^*(\nu)}(a|s)}{\pi_{\theta^K(\nu)}(a|s)}\right) \\
&= \mathbb{E}_{\rho(s)}[D_f(\pi_{\theta^*(\nu)}(a|s), \pi_{\theta^K(\nu)}(a|s))],
\end{aligned}
\tag{53}
$$

where, in the first equation we expand upon the sum over all trajectories with the occupancy distribution over states and actions, and replacing with f-divergence, we get the final expression.

Next, we expand similarly for the term II in (51) and expand as

$$
\begin{aligned}
D_f(P(\tau;\beta), P(\tau;\theta^K(\nu))) & \\
&= \sum_\tau P(\tau;\theta^K(\nu))) f\left(\frac{P(\tau;\beta)}{P(\tau;\theta^K(\nu))}\right) \\
&= \sum_\tau P(\tau;\theta^K(\nu))) f\left(\frac{\rho(s_0)\pi_{\theta^K(\nu)}(a_1|s_0)\mathbb{P}(s_2|s_1,a_1)\prod_{h=2}^H \pi_{\theta^*(\nu)}(a_h|s_h)\mathbb{P}(s_{h+1}|s_h,a_h)}{\rho_0(s_0)\pi_{\theta^K(\nu)}(a_1|s_0)\mathbb{P}(s_2|s_1,a_1)\prod_{h=2}^H \pi_{\theta^K(\nu)}(a_h|s_h)\mathbb{P}(s_{h+1}|s_h,a_h)}\right) \\
&= \sum_{s_0} \rho(s_0) \sum_{a_1} \pi_{\theta^K(\nu)}(a_1|s_0) \sum_{s_1} \mathbb{P}(s_1|s_0,a_0) \cdots f\left(\frac{\prod_{h=2}^H \pi_{\theta^*(\nu)}(a_h|s_h)\mathbb{P}(s_{h+1}|s_h,a_h)}{\prod_{h=2}^H \pi_{\theta^K(\nu)}(a_h|s_h)\mathbb{P}(s_{h+1}|s_h,a_h)}\right) \\
&= \sum_{s_0} \rho(s_0) \sum_{a_1} \pi_{\theta^K(\nu)}(a_1|s_0) \sum_{\tau_1} P(\tau_1;\theta^K(\nu), \frac{P(\tau_1;\theta^*(\nu)}{P(\tau_1;\theta^K(\nu))} \\
&= \sum_{s_0} \rho(s_0) \sum_{a_1} \pi_{\theta^K(\nu)}(a_1|s_0) D_f(P(\tau_1;\theta^*(\nu), P(\tau_1;\theta^K(\nu)),
\end{aligned}
\tag{54}
$$

where we expand the trajectory distribution induced by the policies and subsequently express as the ratio of the probability of trajectories wrt $\tau_1$, we get the final expression. Now, we expand upon the f-divergence of the trajectory $\tau_1$ distribution as

$$
\begin{aligned}
D_f(P(\tau;\beta), P(\tau;\theta^K(\nu))) &= \sum_{s_0} \rho(s_0) \sum_{a_1} \pi_{\theta^K(\nu)}(a_1|s_0) D_f(P(\tau_1;\theta^*(\nu), P(\tau_1;\theta^K(\nu)) \\
&\leq \sum_{s_0} \rho(s_0) \sum_{a_1} \pi_{\theta^K(\nu)}(a_1|s_0) \Big( D_f(P(\tau_1;\theta^*(\nu)), P(\tau_1;\beta)) \\
&\quad + D_f(P(\tau_1;\beta), P(\tau_1;\theta^K(\nu))) \Big),
\end{aligned}
\tag{55}
$$

where using the triangle inequality and get back the similar form with which we had started in equation (51) similar to term I and term II. Here, similarly continuing this expansion, we finally get

$$
\begin{aligned}
D_f(P(\tau; \theta^*(\nu)), P(\tau; \theta^K(\nu))) &\leq \sum_{h=0}^{H-1} \mathbb{E}_{s \sim \rho_{\theta^K(\nu)}^{\mathbb{P}}(s)} D_f(\pi_{\theta^*(\nu)}(a|s), \pi_{\theta^K(\nu)}(a|s)) \\
&\leq H \mathbb{E}_{s \sim \rho_{\theta^K(\nu)}(s)} D_f(\pi_{\theta^*(\nu)}(a|s), \pi_{\theta^K(\nu)}(a|s)) \\
&\leq H \sum_s \rho_{\theta^K(\nu)}(s) D_f(\pi_{\theta^*(\nu)}(a|s), \pi_{\theta^K(\nu)}(a|s)) \\
&\leq H D_f(\pi_{\theta^*(\nu)}(a'|s'), \pi_{\theta^K(\nu)}(a'|s')),
\end{aligned}
\tag{56}
$$

where, we upper bound the first equation by the total number of timesteps or horizon length $H$ of the trajectory and subsequently upper-bound the divergence by the state $(s, a)$ pair for which the $D_f(\pi_{\theta^*(\nu)}(a|s), \pi_{\theta^K(\nu)}(a|s))$ is maximum and is given as $(s', a')$. Next, in (56), by considering the total variation as the f-divergence and expanding using definition with countable measures to obtain

$$
\begin{aligned}
D_f(P(\tau; \theta^*(\nu)), P(\tau; \theta^K(\nu))) &\leq H D_{TV}(\pi_{\theta^*(\nu)}(a|s'), \pi_{\theta^K(\nu)}(a|s')) \\
&\leq \frac{H}{2} \|\pi_{\theta^*(\nu)}(a|s) - \pi_{\theta^K(\nu)}(a|s)\|_1 \\
&\leq \frac{H L_2}{2} \|\theta^*(\nu) - \theta^K(\nu)\|,
\end{aligned}
\tag{57}
$$

where, we use the Lipschitz assumption (cf. Assumption 3) on the policy parameter to get the final expression of equation (57). We note that the result holds for any general horizon length $H$. $\qquad \square$

## 11.2. Proof of Lemma 4.2

### 11.2.1. PROOF OF LEMMA 4.2 STATEMENT (I)

*Proof.* We start with the definition from (29)

$$
\begin{aligned}
\nabla_{\nu,\theta}^2 V_s(\nu_t, \theta^K(\nu_t)) &= \sum_\tau P(\tau; \theta^K(\nu_t)) \left[ \sum_{h=0}^{H_\ell} \gamma^h \cdot \nabla_\nu r_{\nu_t}(s_h, a_h) \cdot \left( \sum_{j=0}^{H_\ell} \nabla_\theta \log \pi_{\theta^K(\nu_t)}(a_j|s_j) \right)^T \right] \\
&\leq \left[ \sum_{h=0}^{H_\ell} \gamma^h \cdot \nabla_\nu r_{\nu_t}(s_h', a_h') \cdot \left( \sum_{j=0}^{H_\ell} \nabla_\theta \log \pi_{\theta^K(\nu_t)}(a_j'|s_j') \right)^T \right] \sum_\tau P(\tau; \theta^K(\nu_t)) \\
&= \left[ \sum_{h=0}^{H_\ell} \gamma^h \cdot \nabla_\nu r_{\nu_t}(s_h', a_h') \cdot \left( \sum_{j=0}^{H_\ell} \nabla_\theta \log \pi_{\theta^K(\nu_t)}(a_j'|s_j') \right)^T \right],
\end{aligned}
\tag{58}
$$

where $\tau'$ represents the trajectory for which the inner product term is maximum and thereby upper-bounding with that leads to expression in equation (58) Next we upper-bound the norm of the $\|\nabla_{\nu,\theta}^2 V_s(\nu_t, \theta^K(\nu_t))\|$ as

$$
\begin{aligned}
\|\nabla_{\nu,\theta}^2 V_s(\nu_t, \theta^K(\nu_t))\| &\leq \left\| \sum_{h=0}^{H_\ell} \gamma^h \cdot \nabla_\nu r_{\nu_t}(s_h', a_h') \cdot \left( \sum_{j=0}^{H_\ell} \nabla_\theta \log \pi_{\theta^K(\nu_t)}(a_j'|s_j') \right)^T \right\| \\
&\leq \left\| \sum_{h=0}^{H_\ell} \gamma^h \cdot \nabla_\nu r_{\nu_t}(s_h', a_h') \right\| \cdot \left\| \sum_{j=0}^{H_\ell} \nabla_\theta \log \pi_{\theta^K(\nu_t)}(a_j'|s_j') \right\| \\
&\leq H_\ell^2 L_r B,
\end{aligned}
\tag{59}
$$

where we apply the Cauchy-Schwarz inequality to get the equation in the second line. We apply triangle inequality with Assumptions 2 and 3 to get the final bound for the mixed hessian term. $\qquad \square$

11.2.2. PROOF OF LEMMA 4.2 STATEMENT (II)

*Proof.* We start by considering the term

$$\nabla_\theta^2 V_s(\nu, \theta^*(\nu)) - \nabla_\theta^2 V_s(\nu, \theta^K(\nu)) = \mathbb{E}_{P(\tau;\theta^*(\nu))} f_1(\tau) - \mathbb{E}_{P(\tau;\theta^K(\nu))} f_2(\tau), \tag{60}$$

where we define

$$f_1(\tau) = \sum_{h=0}^{H_\ell} \gamma^{h-1} \cdot r_{\nu_t}(s_h, a_h) \cdot \left( \sum_{j=0}^{H_\ell} \nabla_\theta^2 \log \pi_{\theta^*(\nu)}(a_j|s_j) \right) \tag{61}$$

$$f_2(\tau) = \sum_{h=0}^{H_\ell} \gamma^{h-1} \cdot r_{\nu_t}(s_k, a_k) \cdot \left( \sum_{j=0}^{H_\ell} \nabla_\theta^2 \log \pi_{\theta^K(\nu)}(a_j|s_j) \right). \tag{62}$$

Subsequently, we write the norm of the equation (60) into 2 parts as

$$
\begin{aligned}
&\|\nabla_\theta^2 V_s(\nu, \theta^*(\nu)) - \nabla_\theta^2 V_s(\nu, \theta^K(\nu))\| \\
&= \|\mathbb{E}_{P(\tau;\theta^*(\nu))} f_1(\tau) - \mathbb{E}_{P(\tau;\theta^K(\nu))} f_1(\tau) + \mathbb{E}_{P(\tau;\theta^K(\nu))} f_1(\tau) - \mathbb{E}_{P(\tau;\theta^K(\nu))} f_2(\tau)\| \\
&\leq \|T_1\| + \|T_2\|,
\end{aligned} \tag{63}
$$

where, $T_1 = \mathbb{E}_{P(\tau;\theta^*(\nu))} f_1(\tau) - \mathbb{E}_{P(\tau;\theta^K(\nu))} f_1(\tau)$ and $T_2 = \mathbb{E}_{P(\tau;\theta^K(\nu))} f_1(\tau) - \mathbb{E}_{P(\tau;\theta^K(\nu))} f_2(\tau)$. We next, upper-bound the individual terms to get the final Lispchitz constant.

First, we focus on the first term of inequality i.e $T_1$ given as

$$
\begin{aligned}
T_1 &= \mathbb{E}_{P(\tau;\theta^*(\nu))} f_1(\tau) - \mathbb{E}_{P(\tau;\theta^K(\nu))} f_1(\tau) \\
&\leq \sup_{f_1} [\mathbb{E}_{P(\tau;\theta^*(\nu))} f_1(\tau) - \mathbb{E}_{P(\tau;\theta^K(\nu))} f_1(\tau)] \\
&\leq L_{f_1} \chi_1 d_{TV}(P(\tau;\theta^*(\nu)), P(\tau;\theta^K(\nu))) \\
&\leq L_{f_1} \chi_1 \frac{H_\ell}{2} L_2 \|\theta^*(\nu) - \theta^K(\nu)\|,
\end{aligned} \tag{64}
$$

where we convert the inequality first to a standard Integral Probability Metric form by taking the supremum and then dividing and multiplying with the Lipschitz constant $L_{f_1}$ from equation (78), we get the final expression in terms of Total variation where $\chi_1$ is the constant. Then, we upper-bounded the total variation using the results from equation (57) to get the final expression.

Now, we proceed to the second term of the equation $T_2$ and derive an upper bound as

$$
\begin{aligned}
T_2 &= \sum_\tau P(\tau;\theta^K(\nu))(f_1(\tau) - f_2(\tau)) \\
&\leq f_1(\tau') - f_2(\tau') \\
&= \sum_{h=0}^{H_\ell} \gamma^{h-1} \cdot r_{\nu_t}(s_h, a_h) \cdot \left( \sum_{j=0}^{H_\ell} (\nabla_\theta^2 \log \pi_{\theta^*(\nu)}(a_j|s_j) - \nabla_\theta^2 \log \pi_{\theta^K(\nu)}(a_j|s_j)) \right)
\end{aligned} \tag{65}
$$

,

where we consider the trajectory $\tau'$ in the sum with the maximum value and upper bound by that to get equation (65).

$$
\begin{aligned}
\|T_2\| &\leq \|\sum_{h=0}^{H_\ell} \gamma^{h-1} \cdot r_{\nu_t}(s_h, a_h) \cdot \left(\sum_{j=0}^{H_\ell} (\nabla_\theta^2 \log \pi_{\theta^*(\nu)}(a_j|s_j) - \nabla_\theta^2 \log \pi_{\theta^K(\nu)}(a_j|s_j))\right)\| \qquad (66) \\
&\leq \|\sum_{h=0}^{H_\ell} \gamma^{h-1} \cdot r_{\nu_t}(s_h, a_h) \cdot \left(\sum_{j=0}^{H_\ell} (\nabla_\theta^2 \log \pi_{\theta^*(\nu)}(a_j|s_j) - \nabla_\theta^2 \log \pi_{\theta^K(\nu)}(a_j|s_j))\right)\| \\
&\leq \sum_{h=0}^{H_\ell} \gamma^{h-1} \cdot \|r_{\nu_t}(s_h, a_h)\| \left(\sum_{j=0}^{H_\ell} (\|\nabla_\theta^2 \log \pi_{\theta^*(\nu)}(a_j|s_j) - \nabla_\theta^2 \log \pi_{\theta^K(\nu)}(a_j|s_j)\|)\right) \\
&\leq \sum_{h=0}^{H_\ell} \gamma^{h-1} \cdot \|r_{\nu_t}(s_h, a_h)\| H_\ell L_2 \|\theta^*(\nu) - \theta^K(\nu)\| \\
&\leq L_2 R H_\ell \|\theta^*(\nu) - \theta^K(\nu)\| \sum_{h=0}^{H_\ell} \gamma^{h-1} \\
&= L_2 R H_\ell^2 \|\theta^*(\nu) - \theta^K(\nu)\|,
\end{aligned}
$$

where we use Cauchy-Schwartz and triangle inequality repetitively to get to the third inequality. Next, we use Assumption 3 on the Lipschitzness of the gradient of the score function and the bounded reward norm $\max_{(s,a)} \|r_\nu(s,a)\| = R$ to get the next inequality. Finally, we use the upper bound on the geometric series to obtain the final expression.

Adding equations (64) and (66), we get the

$$
\|\nabla_\theta^2 V_s(\nu, \theta^*(\nu)) - \nabla_\theta^2 V_s(\nu, \theta^K(\nu))\| \leq L' \|\theta^*(\nu) - \theta^K(\nu)\| \qquad (67)
$$

where, $L' = L_{f_1} \chi_1 \frac{H_\ell}{2} L_2 + L_2 R H_\ell^2$ and $L_{f_1} = L_2 H_\ell^2 R$. $\qquad \square$

### 11.2.3. PROOF OF LEMMA 4.2 STATEMENT (III)

*Proof.* We start by considering the term

$$
\nabla_{\nu,\theta}^2 V_s(\nu, \theta^*(\nu)) - \nabla_{\nu,\theta}^2 V_s(\nu, \theta^K(\nu)) \leq \mathbb{E}_{P(\tau;\theta^*(\nu))} f_3(\tau) - \mathbb{E}_{P(\tau;\theta^K(\nu))} f_4(\tau) \qquad (68)
$$

where we define

$$
f_3(\tau) = \sum_{h=0}^{H_\ell} \gamma^{h-1} \cdot \nabla_\nu r_\nu(s_h, a_h) \cdot \left(\sum_{j=0}^{H_\ell} \nabla_\theta \log \pi_{\theta^*(\nu)}(a_j|s_j)\right)^T \qquad (69)
$$

$$
f_4(\tau) = \sum_{h=0}^{H_\ell} \gamma^{h-1} \cdot \nabla_\nu r_\nu(s_h, a_h) \cdot \left(\sum_{j=0}^{H_\ell} \nabla_\theta \log \pi_{\theta^K(\nu)}(a_j|s_j)\right)^T. \qquad (70)
$$

Subsequently, we write the norm of the equation (68) into 2 parts as

$$
\begin{aligned}
\|\nabla_{\nu,\theta}^2 V_s(\nu, \theta^*(\nu)) - \nabla_{\nu,\theta}^2 V_s(\nu, \theta^K(\nu))\| &\leq \|\mathbb{E}_{P(\tau;\theta^*(\nu))} f_3(\tau) - \mathbb{E}_{P(\tau;\theta^K(\nu))} f_3(\tau) \qquad (71) \\
&\quad + \mathbb{E}_{P(\tau;\theta^K(\nu))} f_3(\tau) - \mathbb{E}_{P(\tau;\theta^K(\nu))} f_4(\tau)\| \\
&\leq \|T_3\| + \|T_4\|,
\end{aligned}
$$

where, $T_3 = \mathbb{E}_{P(\tau;\theta^*(\nu))} f_3(\tau) - \mathbb{E}_{P(\tau;\theta^K(\nu))} f_3(\tau)$ and $T_4 = \mathbb{E}_{P(\tau;\theta^K(\nu))} f_3(\tau) - \mathbb{E}_{P(\tau;\theta^K(\nu))} f_4(\tau)$. We next, upper-bound the individual terms to get the final Lispchitz constant.

First, we focus on the first term of inequality i.e $T_3$ given as

$$
\begin{aligned}
T_3 &= \mathbb{E}_{P(\tau;\theta^*(\nu))} f_3(\tau) - \mathbb{E}_{P(\tau;\theta^K(\nu))} f_3(\tau) \\
&\leq \sup_{f_3}[\mathbb{E}_{P(\tau;\theta^*(\nu))} f_3(\tau) - \mathbb{E}_{P(\tau;\theta^K(\nu))} f_1(\tau)] \\
&\leq L_{f_3}\chi_2 d_{TV}(P(\tau;\theta^*(\nu)), P(\tau;\theta^K(\nu))) \\
&\leq L_{f_3}\chi_2 \frac{H_\ell}{2} L_2 \|\theta^*(\nu) - \theta^K(\nu)\|,
\end{aligned}
\tag{72}
$$

where we convert the inequality first to a standard Integral Probability Metric form by taking the supremum and then dividing and multiplying with the Lipschitz constant $L_{f_3}$ from equation (81), and then we get the final expression in terms of Total variation where $\chi_2$ is the constant. Then, we upper-bounded the total variation using the results from equation (57) to get the final expression.

Now, we proceed to the second term of the equation $T_4$ and derive an upper bound as

$$
\begin{aligned}
T_4 &= \sum_\tau P(\tau;\theta^K(\nu))(f_3(\tau) - f_4(\tau)) \\
&\leq f_3(\tau') - f_4(\tau') \\
&= \sum_{h=0}^{H_\ell} \gamma^{h-1} \cdot \nabla_\nu r_\nu(s_h, a_h) \cdot \left( \sum_{j=0}^{H_\ell} (\nabla_\theta \log \pi_{\theta^*(\nu)}(a_j|s_j) - \nabla_\theta \log \pi_{\theta^K(\nu)}(a_j|s_j)) \right)^T
\end{aligned}
\tag{73}
$$

where we consider the trajectory $\tau'$ in the sum with the maximum value and upper bound by that to get equation (73).

Next, we upper-bound the norm $\|T_4\|$ as

$$
\begin{aligned}
\|T_4\| &\leq \left\| \sum_{h=0}^{H_\ell} \gamma^{h-1} \cdot \nabla_\nu r_\nu(s_h, a_h) \cdot \left( \sum_{j=0}^{H_\ell} (\nabla_\theta \log \pi_{\theta^*(\nu)}(a_j|s_j) - \nabla_\theta \log \pi_{\theta^K(\nu)}(a_j|s_j)) \right)^T \right\| \\
&\leq \sum_{h=0}^{H_\ell} \gamma^{h-1} \cdot \|\nabla_\nu r_\nu(s_h, a_h)\| \left( \sum_{j=0}^{H_\ell} \|\nabla_\theta \log \pi_{\theta^*(\nu)}(a_j|s_j) - \nabla_\theta \log \pi_{\theta^K(\nu)}(a_j|s_j)\| \right) \\
&\leq \sum_{h=0}^{H_\ell} \gamma^{h-1} \cdot \|\nabla_\nu r_\nu(s_h, a_h)\| H_\ell L_1 \|\theta^*(\nu) - \theta^K(\nu)\| \\
&\leq L_1 L_r H_\ell \|\theta^*(\nu) - \theta^K(\nu)\| \sum_{h=0}^{H_h} \gamma^{h-1} \\
&\leq L_1 L_r H_\ell^2 \|\theta^*(\nu) - \theta^K(\nu)\|,
\end{aligned}
\tag{74}
$$

where we use Cauchy-Schwartz and triangle inequality repetitively to get to the third inequality. Next, we use Assumption 3 on the Lipschitzness of the gradient of the score function and the bounded reward $R$ (cf. Assumption 2). Finally, we upper-bound sum of this geometric series to obtain the final expression. Adding equations (72) and (74), we get the

$$
\|\nabla^2_{\nu,\theta} V_s(\nu, \theta^*(\nu_t)) - \nabla^2_{\nu,\theta} V_s(\nu_t, \theta^K(\nu))\| \leq L'' \|\theta^*(\nu) - \theta^K(\nu)\|
\tag{75}
$$

where, $L'' = L_{f_3}\chi_2 \frac{H_\ell}{2} L_2 + L_1 L_r H_\ell^2$ and $L_{f_3} = L_r L_1 H_\ell^2$. $\qquad\square$

## 12. Additional Supporting Lemmas

### 12.1. Proof of Lispchitzness for $f_1(\cdot)$ defined in (61)

Here, we prove that the function denoted as $f_1(\theta^*(\nu)) = \sum_{h=0}^{H_\ell} \gamma^{h-1} \cdot r_{\nu_t}(s_h, a_h) \cdot \left( \sum_{j=0}^{H_\ell} \nabla_\theta^2 \log \pi_{\theta^*(\nu)}(a_j|s_j) \right)$ is Lispchitz continuous w.r.t $\theta$ with Lipschitz constant $L_{f_1}$ i.e

$$\|f_1(\theta^*(\nu)) - f_1(\theta^K(\nu))\| \le L_{f_1} \|\theta^*(\nu) - \theta^K(\nu)\|. \tag{76}$$

First, we begin with the difference term as :

$$f_1(\theta^*(\nu)) - f_1(\theta^K(\nu)) = \sum_{h=0}^{H_\ell} \gamma^{h-1} r_{\nu_t}(s_h, a_h) \left( \sum_{j=0}^{H_\ell} (\nabla_\theta^2 \log \pi_{\theta^*(\nu)}(a_j|s_j) - \nabla_\theta^2 \log \pi_{\theta^K(\nu)}(a_j|s_j)) \right). \tag{77}$$

Subsequently, taking the norm we get

$$
\begin{aligned}
&\|f_1(\theta^*(\nu)) - f_1(\theta^K(\nu))\| \\
&= \left\| \sum_{h=0}^{H_\ell} \gamma^{h-1} r_{\nu_t}(s_h, a_h) \left( \sum_{j=0}^{H_\ell} (\nabla_\theta^2 \log \pi_{\theta^*(\nu)}(a_j|s_j) - \nabla_\theta^2 \log \pi_{\theta^K(\nu)}(a_j|s_j)) \right) \right\| \\
&\le \sum_{h=0}^{H_\ell} \gamma^{h-1} \cdot \|r_{\nu_t}(s_h, a_h)\| \cdot \left( \sum_{j=0}^{H_\ell} \|\nabla_\theta^2 \log \pi_{\theta^*(\nu)}(a_j|s_j) - \nabla_\theta^2 \log \pi_{\theta^K(\nu)}(a_j|s_j)\| \right) \\
&\le \sum_{h=0}^{H_\ell} \gamma^{h-1} \cdot \|r_{\nu_t}(s_h, a_h)\| L_2 H_\ell \|\theta^*(\nu) - \theta^K(\nu)\| \\
&\le L_2 H_\ell^2 R \|\theta^*(\nu) - \theta^K(\nu)\|,
\end{aligned}
\tag{78}
$$

where we use Cauchy-Schwartz and triangle inequality repeatedly to get the subsequent inequalities. In the third inequality, we use the Lipschitzness assumption of $\nabla_\theta^2 \log \pi_{\theta^*(\nu)}(a_j|s_j)$ from Assumption 3 and finally using the boundedness of the reward values and upper-bounding the Geometric series, we get the final expression. Thus $f_1(\theta^*(\nu))$ is Lispchitz continuous w.r.t $\theta$ with Lipschitz constant $L_{f_1} = L_2 H_\ell^2 R$.

### 12.2. Proof of Lispchitzness for $f_3(\cdot)$ defined in (69)

Here, we prove that the function denoted as $f_3(\theta^*(\nu)) = \sum_{h=0}^{H_\ell} \gamma^{h-1} \cdot \nabla_\nu r_\nu(s_h, a_h) \cdot \left( \sum_{j=0}^{H_\ell} \nabla_\theta \log \pi_{\theta^*(\nu)}(a_j|s_j) \right)^T$ is Lispchitz continuous w.r.t $\theta$ with Lipschitz constant $L_{f_3}$ i.e

$$\|f_3(\theta^*(\nu)) - f_3(\theta^K(\nu))\| \le L_{f_3} \|\theta^*(\nu) - \theta^K(\nu)\| \tag{79}$$

First, we begin with the difference term as :

$$
\begin{aligned}
&f_3(\theta^*(\nu)) - f_3(\theta^K(\nu)) \\
&= \sum_{h=0}^{H_\ell} \gamma^{h-1} \cdot \nabla_\nu r_\nu(s_h, a_h) \cdot \left( \sum_{j=0}^{H_\ell} (\nabla_\theta \log \pi_{\theta^*(\nu)}(a_j|s_j) - \nabla_\theta \log \pi_{\theta^*(\nu)}(a_j|s_j)) \right)^T
\end{aligned}
\tag{80}
$$

Subsequently, taking the norm we get

$$\|f_3(\theta^*(\nu)) - f_3(\theta^K(\nu))\|$$

$$= \| \sum_{h=0}^{H_\ell} \gamma^{h-1} \cdot \nabla_\nu r_\nu(s_h, a_h) \cdot \left( \sum_{j=0}^{H_\ell} (\nabla_\theta \log \pi_{\theta^*(\nu)}(a_j|s_j) - \nabla_\theta \log \pi_{\theta^*(\nu)}(a_j|s_j)) \right)^T \|$$

$$\leq \sum_{h=0}^{H_\ell} \gamma^{h-1} \cdot \|\nabla_\nu r_\nu(s_h, a_h)\| \cdot \left( \sum_{j=0}^{H_\ell} (\|\nabla_\theta \log \pi_{\theta^*(\nu)}(a_j|s_j) - \nabla_\theta \log \pi_{\theta^*(\nu)}(a_j|s_j)\|) \right)$$

$$\leq L_r L_1 H_\ell \|\theta^*(\nu) - \theta^K(\nu)\| \sum_{h=0}^{H_\ell} \gamma^{h-1}$$

$$\leq L_r L_1 H_\ell^2 \|\theta^*(\nu) - \theta^K(\nu)\|, \tag{81}$$

where we use Cauchy-Schwartz and triangle inequality repeatedly to get the subsequent inequalities. In the third inequality, we use the Lipschitzness assumption of $\nabla_\theta^2 \log \pi_{\theta^*(\nu)}(a_j|s_j)$ from Assumption 3. Finally, using the Assumption 2 i.e reward Lipschitzness and upper-bounding the Geometric series, we get the final expression. Thus $f_3(\theta^*(\nu))$ is Lispchitz continuous w.r.t $\theta$ with Lipschitz constant $L_{f_3} = L_r L_1 H_\ell^2$.

### 12.3. Proof of Smoothness Condition on the Value function

Here, we prove the smoothness of the value function i.e. the gradient of the value function is Lispchitz continuous w.r.t $\theta$ with Lipschitz constant $L_1$. We begin with the definition of the gradient difference of the value function from the equation (27) as:

$$\nabla_\theta V_s(\nu, \theta^*(\nu)) - \nabla_\theta V_s(\nu_t, \theta^K(\nu)) = \mathbb{E}_{P(\tau;\theta^*(\nu))} V_1(\tau) - \mathbb{E}_{P(\tau;\theta^K(\nu))} V_2(\tau) \tag{82}$$

$$= \mathbb{E}_{P(\tau;\theta^*(\nu))} V_1(\tau) - \mathbb{E}_{P(\tau;\theta^K(\nu))} V_1(\tau)$$

$$+ \mathbb{E}_{P(\tau;\theta^K(\nu))} [V_1(\tau) - V_2(\tau)]$$

$$= \Sigma_1 + \Sigma_2$$

where, first we substitute $V_1 = \sum_{h=0}^{H_\ell-1} \gamma^{h-1} r_{\nu_t}(s_h, a_h) \left( \sum_{j=0}^{H_\ell-1} \nabla_\theta \log \pi_{\theta^*(\nu_t)}(a_j|s_j) \right)$ and $V_2 = \sum_{h=0}^{H_\ell-1} \gamma^{h-1} r_{\nu_t}(s_h, a_h) \left( \sum_{j=0}^{H_\ell-1} \nabla_\theta \log \pi_{\theta^K(\nu_t)}(a_j|s_j) \right)$. Subsequently, by adding and subtracting $\mathbb{E}_{P(\tau;\theta^K(\nu))} V_2(\tau)$, we get the final expression, where $\Sigma_1 = \mathbb{E}_{P(\tau;\theta^*(\nu))} V_1(\tau) - \mathbb{E}_{P(\tau;\theta^K(\nu))} V_1(\tau)$ and $\Sigma_2 = \mathbb{E}_{P(\tau;\theta^K(\nu))} [V_1(\tau) - V_2(\tau)]$. Now, first, we derive the Lipschitz constant for $V_1$ as

$$\|V_1(\theta^*(\nu) - V_1(\theta^K(\nu))\|$$

$$\leq \sum_{h=0}^{H_\ell-1} \gamma^{h-1} \|r_{\nu_t}(s_h, a_h)\| \left( \sum_{j=0}^{H_\ell-1} \|\nabla_\theta \log \pi_{\theta^*(\nu_t)}(a_j|s_j) - \nabla_\theta \log \pi_{\theta^K(\nu_t)}(a_j|s_j)\| \right)$$

$$\leq H_l^2 R L_1 \|\theta^*(\nu) - \theta^K(\nu)\|, \tag{83}$$

where we first use Cauchy-Schwartz and triangle inequality to get the first inequality. Next, we upper-bound the reward with $R$ from Assumption 2, Lipschitzness of policy gradient from Assumption 3, and finally upper-bounding the Geometric series, we get the final expression. The Lipschitz constant $L_5 = H_l^2 R L_1$.

We can subsequently upper-bound $\Sigma_1$ with the total variation distance as

$$\Sigma_1 \leq \sup_V [\mathbb{E}_{P(\tau;\theta^*(\nu))} V(\tau) - \mathbb{E}_{P(\tau;\theta^K(\nu))} V(\tau)] \tag{84}$$

$$\leq L_5 d_{TV}(P(\tau; \theta^K(\nu)), P(\tau; \theta^*(\nu)))$$

$$\leq L_5 \frac{H L_2}{2} \|\theta^*(\nu) - \theta^K(\nu)\|$$

where first we divide by the Lipschitz constant of the function and subsequently upper-bound with the Total Variation distance. Finally, we substitute the total variation distance expression from the equation (57) to get the final expression.

Now, the second term can be written as

$$\Sigma_2 = \mathbb{E}_{P(\tau;\theta^K(\nu))}[V_1(\tau) - V_2(\tau)] \leq V_1(\tau') - V_2(\tau'), \tag{85}$$

where we take the trajectory with the maximum difference and upper-bound the term. Subsequently, $\|\Sigma_2\| \leq \|V_1(\tau') - V_2(\tau')\| = H_l^2 R L_{\theta_1} \|\theta^*(\nu) - \theta^K(\nu)\|$ from equation (83).

Finally, the norm of the gradient difference of the value function from equation (82) as

$$\|\nabla_\theta V_s(\nu, \theta^*(\nu)) - \nabla_\theta V_s(\nu_t, \theta^K(\nu))\| \leq L_6 \|\theta^*(\nu) - \theta^K(\nu)\| \tag{86}$$

where $L_6 = L_5 \frac{HL_2}{2} + L_5$ and $L_5 = H_l^2 R L_{\theta_1}$

### 12.4. Upper-bound on the Norm of the hessian defined in (28)

Here, we prove an upper-bound on the norm of the hessian defined in (28) given as

$$\nabla_\theta^2 V_s(\nu_t, \theta^K(\nu_t)) = \mathbb{E}_{P(\tau;\theta^K(\nu_t))} \left[ \sum_{h=0}^{H_\ell} \gamma^{h-1} r_{\nu_t}(s_h, a_h) \left( \sum_{j=0}^{H_\ell} \nabla_\theta^2 \log \pi_{\theta^K(\nu_t)}(a_j|s_j) \right) \right] \tag{87}$$

$$\leq \sum_{h=0}^{H_\ell} \gamma^{h-1} r_{\nu_t}(s_h, a_h) \left( \sum_{j=0}^{H_\ell} \nabla_\theta^2 \log \pi_{\theta^K(\nu_t)}(a_j|s_j) \right) \sum_\tau P(\tau;\theta^K(\nu_t))$$

$$= \sum_{h=0}^{H_\ell} \gamma^{h-1} r_{\nu_t}(s_h, a_h) \left( \sum_{j=0}^{H_\ell} \nabla_\theta^2 \log \pi_{\theta^K(\nu_t)}(a_j|s_j) \right)$$

where, first we upper-bound the function with the trajectory which has the maximum inner-value. Next we, upper-bound the norm as

$$\|\nabla_\theta^2 V_s(\nu_t, \theta^K(\nu_t))\| \leq \| \sum_{h=0}^{H_\ell} \gamma^{h-1} r_{\nu_t}(s_h, a_h) \left( \sum_{j=0}^{H_\ell} \nabla_\theta^2 \log \pi_{\theta^K(\nu_t)}(a_j|s_j) \right) \| \tag{88}$$

$$\leq \sum_{h=0}^{H_\ell} \gamma^{h-1} \|r_{\nu_t}(s_h, a_h)\| \left( \sum_{j=0}^{H_\ell} \|\nabla_\theta^2 \log \pi_{\theta^K(\nu_t)}(a_j|s_j)\| \right)$$

$$\leq H_l^2 R L_\pi^1$$

where, we first upper-bound with successive application of Cauchy-Schwartz and Triangle inequality to get the second inequality. Finally, with the upper bound on $\|r_{\nu_t}(s_h, a_h)\| \leq R$ from Assumption (15), Lipschitzness of policy gradients from Assumption 2 and upper-bounding the geometric series, we get the final expression.

### 12.5. Upper-bound on the Norm of the hessian defined in (28)

Here, we prove an upper-bound on the norm of the mixed second-order Jaccobian term defined in (29) given as

$$\nabla_{\nu,\theta}^2 V_s(\nu_t, \theta^K(\nu_t)) = \mathbb{E}_{P(\tau;\theta^K(\nu_t))} \left[ \sum_{h=0}^{H_\ell} \gamma^{h-1} \nabla_\nu r_{\nu_t}(s_h, a_h) \left( \sum_{j=0}^{H_\ell} [\nabla_\theta \log \pi_{\theta^K(\nu_t)}(a_j|s_j)]^T \right) \right] \tag{89}$$

$$\leq \sum_{h=0}^{H_\ell} \gamma^{h-1} \nabla_\nu r_{\nu_t}(s_h, a_h) \left( \sum_{j=0}^{H_\ell} [\nabla_\theta \log \pi_{\theta^K(\nu_t)}(a_j|s_j)]^T \right)$$

where first we upper-bound the function with the trajectory which has the maximum inner value. Next, we, upper-bound the norm as

$$\|\nabla^2_{\nu,\theta} V_s(\nu_t, \theta^K(\nu_t))\| \leq \| \leq \sum_{h=0}^{H_\ell} \gamma^{h-1} \nabla_\nu r_{\nu_t}(s_h, a_h) \left( \sum_{j=0}^{H_\ell} [\nabla_\theta \log \pi_{\theta^K(\nu_t)}(a_j|s_j)]^T \right) \| \tag{90}$$

$$\leq \sum_{h=0}^{H_\ell} \gamma^{h-1} \|\nabla_\nu r_{\nu_t}(s_h, a_h)\| \left( \sum_{j=0}^{H_\ell} \|\nabla_\theta \log \pi_{\theta^K(\nu_t)}(a_j|s_j)\| \right) \|$$

$$\leq H_l^2 L_r B$$

where we first upper-bound with the successive application of Cauchy-Schwartz and Triangle inequality to get the second inequality. Finally, with the upper bound on $\|\nabla_\nu r_{\nu_t}(s_h, a_h)\| \leq L_r$ from Lipschitzness Assumption (15), Lipschitzness of policy parametrization from Assumption 2 and upper-bounding the geometric series, we get the final expression.

### 12.6. Proof of Lemma 4.3

*Proof.* Let us start by first deriving the upper bounds for the terms $\phi_1$ and $\phi_2$ as defined in the equation (41) as follows. For $\phi_1(\tau)$, we have

$$\|\phi_1(\tau)\| = \left\| U(\tau) \cdot \sum_{h=0}^{H-1} [-\nabla^2_{v,\theta} V_s(\nu, \theta^*(\nu)) \nabla^2_\theta V_s(\nu, \theta^*(\nu))^{-1} \nabla_\theta f_h(\theta^*(\nu))] \right\|. \tag{91}$$

We define the term $\kappa$ which explains the relative conditioning of the two matrix norms. We define the mixed condition number as

$$\kappa = \frac{\|\nabla^2_{v,\theta} V_s(\nu, \theta^*(\nu))\|}{\|\nabla^2_\theta V_s(\nu, \theta^*(\nu))\|} \leq \frac{H_l L_r B}{l_\pi(1 - \gamma)} \tag{92}$$

Next to upper-bound the second term relating to the difference in $\|\phi_1 - \phi_2\|$ in the equation, we proceed first by upper-bounding the product difference

$$\|\Delta\| = \|\Delta_1 - \Delta_2\| \tag{93}$$

where we define

$$\Delta_1 = \nabla^2_{v,\theta} V_s(\nu, \theta^*(\nu)) \nabla^2_\theta V_s(\nu, \theta^*(\nu))^{-1} \nabla_\theta f_h(\theta^*(\nu)), \tag{94}$$

$$\Delta_2 = \nabla^2_{v,\theta} V_s(\nu, \theta^K(\nu)) \nabla^2_\theta V_s(\nu, \theta^K(\nu))^{-1} \nabla_\theta f_h(\theta^K(\nu)). \tag{95}$$

Also, for simplicity of notations lets take $\psi_1 = \nabla^2_{v,\theta} V_s(\nu, \theta^*(\nu)) \nabla^2_\theta V_s(\nu, \theta^*(\nu))^{-1}$ and $\psi_2 = \nabla^2_{v,\theta} V_s(\nu, \theta^K(\nu)) \nabla^2_\theta V_s(\nu, \theta^K(\nu))^{-1}$, which thus (93) boils down to upper-bounding

$$\Delta = \|\psi_1 f_h(\theta^*(\nu)) - \psi_2 f_h(\theta^K(\nu))\| \tag{96}$$

$$= \|\psi_1 f_h(\theta^*(\nu)) - \psi_1 f_h(\theta^K(\nu)) + \psi_1 f_h(\theta^K(\nu)) - \psi_2 f_h(\theta^K(\nu))\|$$

$$\leq \|\psi_1\| \|f_h(\theta^*(\nu)) - f_h(\theta^K(\nu))\| + \|\psi_1 - \psi_2\| \|f_h(\theta^K(\nu))\|$$

$$\leq \kappa L_1 \|\theta^*(\nu) - \theta^K(\nu)\| + L_2 \|\psi_1 - \psi_2\|,$$

where, first we expand add and subtract the term $\psi_1 f_h(\theta^K(\nu))$, and subsequently by applying Cauchy-Schwartz and triangle inequality, we get to the third inequality. For the final inequality, we apply equation and Lispchitzness Assumptions (15) to get the final expression in equation (96). Next, we focus on upper bounding the second term of the expression specifically $\|\psi_1 - \psi_2\|$

$$\|\psi_1 - \psi_2\| = \|\nabla^2_{v,\theta} V_s(\nu, \theta^*(\nu)) \nabla^2_\theta V_s(\nu, \theta^*(\nu))^{-1} - \nabla^2_{v,\theta} V_s(\nu, \theta^*(\nu)) \nabla^2_\theta V_s(\nu, \theta^K(\nu))^{-1} \tag{97}$$

$$+ \nabla^2_{\nu,\theta} V_s(\nu, \theta^*(\nu)) \nabla^2_\theta V_s(\nu, \theta^K(\nu))^{-1} - \nabla^2_{v,\theta} V_s(\nu, \theta^K(\nu)) \nabla^2_\theta V_s(\nu, \theta^K(\nu))^{-1}\|$$

$$= \|\nabla^2_{\nu,\theta} V_s(\nu, \theta^*(\nu)) \nabla^2_\theta V_s(\nu, \theta^*(\nu))^{-1} - \nabla^2_{v,\theta} V_s(\nu, \theta^*(\nu)) \nabla^2_\theta V_s(\nu, \theta^K(\nu))^{-1}\| \tag{98}$$

$$+ \|\nabla^2_{\nu,\theta} V_s(\nu, \theta^*(\nu)) \nabla^2_\theta V_s(\nu, \theta^K(\nu))^{-1} - \nabla^2_{v,\theta} V_s(\nu, \theta^K(\nu)) \nabla^2_\theta V_s(\nu, \theta^K(\nu))^{-1}\|$$

$$= \Psi_{21} + \Psi_{22},$$

where we expand the definition of $\|\psi_1 - \psi_2\|$ and subsequently apply triangle inequality and Cauchy-Schwartz which then boils to upper-bounding the sum of two terms $\Psi_{21} + \Psi_{22}$. For $\Psi_{21}$, we have

$$
\begin{aligned}
\Psi_{21} &\leq \|\nabla^2_{\nu,\theta} V_s(\nu, \theta^*(\nu))\| \|\nabla^2_\theta V_s(\nu, \theta^*(\nu))^{-1} - \nabla^2_{\nu,\theta} V_s(\theta^K(\nu))^{-1}\| \qquad (99)\\
&\leq L_{\nu,\theta} \|\nabla^2_\theta V_s(\nu, \theta^*(\nu))^{-1} - \nabla^2_\theta V_s(\nu, \theta^K(\nu))^{-1}\|\\
&= L_{\nu,\theta} \|\nabla^2_\theta V_s(\nu, \theta^*(\nu))^{-1}(\nabla^2_\theta V_s(\nu, \theta^*(\nu)) - \nabla^2_\theta V_s(\nu, \theta^K(\nu)))\nabla^2_\theta V_s(\nu, \theta^K(\nu))^{-1}\|\\
&\leq L_{\nu,\theta} \|\nabla^2_\theta V_s(\nu, \theta^*(\nu))\|^{-1} \|(\nabla^2_\theta V_s(\nu, \theta^*(\nu)) - \nabla^2_\theta V_s(\nu, \theta^K(\nu)))\| \|\nabla^2_\theta V_s(\nu, \theta^K(\nu))\|^{-1}\\
&\leq \frac{L_{\nu,\theta} L'}{l_\pi^2} \|\theta^*(\nu) - \theta^K(\nu)\|
\end{aligned}
$$

where we use Cauchy-Schwartz inequality and triangle inequality iteratively to get the final inequality. Next, we use the upper bounds and lower bounds of the hessian and mixed hessian matrices defined in Assumptions to get the final expression. Now, $L_{\nu,\theta} = H_l^2 L_r B$ from equation (90), $L' = L_{f_1} \chi_1 \frac{H_\ell}{2} L_2 + L_2 R H_\ell^2$ and $L_{f_1} = L_2 H_\ell^2 R$ from equation (67). Finally, the second-term $\Psi_{22}$ from equation (97) can be upper-bounded as

$$
\begin{aligned}
\Psi_{22} &\leq \|\nabla^2_{\nu,\theta} V_s(\nu, \theta^*(\nu) - \nabla^2_{\nu,\theta} V_s(\nu, \theta^K(\nu))\| \|\nabla^2_\theta V_s(\nu, \theta^K(\nu))\|^{-1}\\
&\leq \frac{L''}{l_\pi} \|\theta^*(\nu) - \theta^K(\nu)\|, \qquad (100)
\end{aligned}
$$

where, similarly we use triangle inequality with Cauchy-Schwartz to get the final upper-bound of equation (100). Now, $L'' = L_{f_3} \chi_2 \frac{H_\ell}{2} L_2 + L_1 L_r H_\ell^2$ and $L_{f_3} = L_r L_1 H_\ell^2$ from equation (75)

Now, combining equations (99) and (100), we get the final upper-bound of the $\|\psi_1 - \psi_2\|$ in equation (97) as

$$
\|\psi_1 - \psi_2\| \leq \left(\frac{L_{\nu,\theta} L'}{l_\pi^2} + \frac{L''}{l_\pi}\right) \|\theta^*(\nu) - \theta^K(\nu)\|. \qquad (101)
$$

Hence, finally replacing the upper-bound of $\Psi_2$ from equation (101) in equation (96) to obtain the upper-bound on the function difference term $\Delta$ as

$$
\begin{aligned}
\Delta &\leq \kappa L_1 \|\theta^*(\nu) - \theta^K(\nu)\| + \left(\frac{L_{\nu,\theta} L'}{l_\pi^2} + \frac{L''}{l_\pi}\right) \|\theta^*(\nu) - \theta^K(\nu)\| \qquad (102)\\
&= \gamma_1 \|\theta^*(\nu) - \theta^K(\nu)\|,
\end{aligned}
$$

with $\gamma_1 := \kappa L_1 + \frac{L_{\nu,\theta} L'}{l_\pi^2} + \frac{L''}{l_\pi}$ Hence, with the above bounds, we proceed to upper-bound the term II in equation (39) i.e $\|\mathbb{E}_{\tau \sim P(\tau; \theta^K(\nu))}[\phi_1(\tau) - \phi_2(\tau)]\|$ as

$$
\begin{aligned}
\|\mathbb{E}_{\tau \sim P(\tau; \theta^K(\nu))}[\phi_1(\tau) - \phi_2(\tau)]\| &\leq \|\phi_1(\tau') - \phi_1(\tau')\| \qquad (103)\\
&= \|U(\tau) \cdot \sum_{h=0}^{H-1}(\Delta_1 - \Delta_2)\|\\
&\leq \|\sum_{h'=0}^{H-1} u(s_h, a_h)\| \|\sum_{h=0}^{H-1}(\Delta_1 - \Delta_2)\|\\
&\leq H^2 \tilde{u} \|\Delta_1 - \Delta_2\|\\
&\leq H^2 \tilde{u} \gamma_1 \|\theta^*(\nu) - \theta^K(\nu)\|,
\end{aligned}
$$

where first we select the trajectory $\tau'$ with the maximum sum and subsequently using the Cauchy-Schwartz inequality we get the second equation. Based on the assumption of bounded utility $u(s, a) \leq \tilde{u}, \forall(s, a)$, we get the third equation and finally using the upper bound of $\Delta_1 - \Delta_2$ from equation (102), we get the final expression for equation (103). $\qquad \square$

**12.7. Proof of Lemma 4.4**

*Proof.* Here, we derive an upper bound on the tracking term due to the use of surrogate gradients $\|\theta^*(\nu_t) - \theta^K(\nu_t)\|$. To begin the proof, we start with the smoothness in the value function shown in equation (86), as

$$V_s(\nu_t, \theta^{k+1}(\nu_t)) \leq V_s(\nu_t, \theta^k(\nu_t)) + \langle \nabla_\theta V_s(\nu_t, \theta^{k+1}(\nu_t)), \theta^{k+1}(\nu_t) - \theta^k(\nu_t) \rangle \tag{104}$$
$$+ \frac{L_6}{2} \|\theta^{k+1}(\nu_t) - \theta^k(\nu_t)\|^2.$$

where $L_6 = L_5 \frac{H_\ell^2 L_2}{2} + L_5$ and $L_5 = H_l R L_{\theta_1}$. Now, from the update of $\theta$ from Algorithm 2, we know that

$$\theta^{k+1}(\nu_t) = \theta^k(\nu_t) - \alpha_\ell \nabla_\theta V_s(\nu_t, \theta^k(\nu_t)). \tag{105}$$

Replacing the update in equation (104), we have

$$V_s(\nu_t, \theta^{k+1}(\nu_t)) \leq V_s(\nu_t, \theta^k(\nu_t)) + \langle \nabla_\theta V_s(\nu_t, \theta^{k+1}(\nu_t)), -\alpha_\ell \nabla_\theta V_s(\nu_t, \theta^k(\nu_t)) \rangle$$
$$+ \frac{L_6}{2} \| - \alpha_\ell \nabla_\theta V_s(\nu_t, \theta^k(\nu_t))\|^2$$
$$= V_s(\nu_t, \theta^k(\nu_t)) - \alpha_\ell \left(1 - \frac{\alpha_\ell L_6}{2}\right) \|\nabla_\theta V_s(\nu_t, \theta^k(\nu_t))\|^2, \tag{106}$$

where we expand the expression after replacing the update in equation (104).

Next, from Assumption 4, we note that for the value function, it holds that

$$\|\nabla_\theta V_s(\nu, \theta^k(\nu))\|^2 \geq \frac{\mu}{2}(V_s(\nu, \theta^k(\nu)) - V_s(\nu, \theta^*(\nu))). \tag{107}$$

Assumption 4 ensures that the objective satisfies the gradient dominance or the PL condition, which can be satisfied in practice for various settings. For instance, Assumption 4 is satisfied in our setting for softmax policy parametrization as detailed in (Mei et al., 2020b, Lemma 8). Now, replacing the PL condition in equation (106), we have

$$V_s(\nu_t, \theta^{k+1}(\nu_t)) - V_s(\nu_t, \theta^k(\nu_t)) \leq -\alpha_\ell(1 - \frac{\alpha_\ell L_6}{2})\frac{\mu}{2}(V_s(\nu, \theta^k(\nu)) - V_s(\nu, \theta^*(\nu)))$$
$$= -\alpha_3(V_s(\nu_t, \theta^k(\nu_t)) - V_s(\nu_t, \theta^*(\nu_t))), \tag{108}$$

where, after replacing the PL condition in equation (106), we substitute $\alpha_3 = \alpha_\ell(1 - \frac{\alpha_\ell L_6}{2})\frac{\mu}{2}$ for simplicity of calculations.

$$V_s(\nu_t, \theta^{k+1}(\nu_t)) - V_s(\nu_t, \theta^*(\nu_t))) \leq (1 - \alpha_3)(V_s(\nu_t, \theta^k(\nu_t)) - V_s(\nu_t, \theta^*(\nu_t)))$$
$$V_s(\nu_t, \theta^K(\nu_t)) - V_s(\nu_t, \theta^*(\nu_t))) \leq (1 - \alpha_3)^K(V_s(\nu_t, \theta^0(\nu_t)) - V_s(\nu_t, \theta^*(\nu_t))), \tag{109}$$

where the first equation comes from algebraic manipulation and applying the equation recursively, we get the second inequality, assuming $0 \leq \alpha_3 \leq 1$. Now, we note that from the smoothness of value function, we have the upper bound

$$V_s(\nu_t, \theta^0(\nu_t)) - V_s(\nu_t, \theta^*(\nu_t)) \leq \frac{L_6}{2} \|\theta^*(\nu_t) - \theta^0(\nu_t)\|^2, \tag{110}$$

where $L_6 = L_5 \frac{H L_2}{2} + L_5$, $L_5 = H_l^2 R L_1$ and we use the Lipschitz smoothness assumption and expand along the point $\nu_t, \theta^*$, for which the gradient term vanishes. Also, since PL implies quadratic growth, it holds that

$$V_s(\nu_t, \theta^K(\nu_t)) - V_s(\nu_t, \theta^*(\nu_t)) \geq \frac{\mu}{2} \|\theta^K(\nu_t) - \theta^*(\nu_t)\|^2. \tag{111}$$

Now, substituting the equations (110), (111) in (109) to obtain

$$\|\theta^K(\nu_t) - \theta^*(\nu)\|^2 \leq (1 - \alpha_3)^K \frac{L_6}{\mu} Z, \tag{112}$$

where, $Z := \max_\nu \|\theta^0 - \theta^*(\nu)\|^2$, $\alpha_3 = \alpha_\ell(1 - \frac{\alpha_\ell L_6}{2})\frac{\mu}{2}$ and $L_6 = L_5 \frac{H_\ell L_2}{2} + L_5$ and $L_5 = H_l^2 R L_1$. $\square$

## 13. Discuss of Assumption 3 and Assumption 4

- Assumption 3 ensure certain properties of the policy parametrization such as Lipschitz policy, bounded score function, Lipschitz score function, and Lipschitz Hessian of the log of the policy. We remark that these assumptions are not restrictive and satisfied in practice for practical classes of policies. For example, this assumption is satisfied for softmax policy parametrization.

$$\pi_\theta(a|s) = \frac{\exp \theta^T \phi(s,a)}{\sum_{a'} \exp \theta^T \phi(s,a')} \tag{113}$$

where, first we write down the expression of the softmax policy gradient parametrization with function approximation $\phi$. Subsequently, taking log and taking the gradient w.r.t to the policy parameterization, we get

$$\nabla_\theta \log \pi_\theta(a|s) = \phi(s,a) - \frac{1}{\sum_{a'} \exp \theta^T \phi(s,a')} \sum_{a'} \exp \theta^T \phi(s,a') \phi(s,a') \tag{114}$$

$$= \phi(s,a) - \sum_{a'} \pi_\theta(a|s) \phi(s,a')$$

where we can denote $\widehat{\phi}_{s,a'} = \sum_{a'} \pi_\theta(a|s)\phi(s,a')$. Now, taking the norm on the LHS of equation (114) we get

$$\|\nabla_\theta \log \pi_\theta(a|s)\| \le \|\phi(s,a)\| + \|\phi(s,a')\| \tag{115}$$

where, we apply triangle inequality to get the final bound, which imposes certain constraints on the norm of the function approximation $\|\phi(s,a)\| \le \zeta_1$ which is a common assumption in various scenarios.

$$\|\nabla_\theta \pi_\theta(a|s)\| \le k\zeta_1 \tag{116}$$

where, we expand $\nabla_\theta \log \pi_\theta(a|s) = \frac{1}{\pi_\theta(a|s)} \nabla_\theta \pi_\theta(a|s)$

- Assumption 4 ensures that the objective function satisfies some geometric properties such as PL condition. We remark that the value function satisfies PL condition with softmax policy parametrization (see (Mei et al., 2020b, Lemma 8)). Further, a property that we need for our analysis to hold is that the Hessian of the objective function has all non-zero eigenvalues. This assumption holds for softmax parametrizations. Now, in order to show that we first consider the softmax-parametrization considered in equation (114), we first compute the hessian as

$$\nabla_\theta^2 \log \pi_\theta(a|s) = \nabla_\theta[\phi(s,a) - \sum_{a'} \pi_\theta(a|s)\phi(s,a')] \tag{117}$$

$$= -\sum_{a'} \nabla_\theta \pi_\theta(a|s)\phi(s,a')^T$$

$$= -\mathbb{E}_\pi[\nabla_\theta \log \pi_\theta(a|s)\phi(s,a')^T]$$

$$= \mathbb{E}_\pi[\widehat{\phi}(s,a')\phi(s,a')^T - \phi(s,a)\phi(s,a')^T]$$

Now, finally, we substitute this to the equation of hessian of the value function in equation (117)

$$\nabla_\theta^2 V_s(\nu_t, \theta^K(\nu_t)) = \mathbb{E}_{P(\tau;\theta)}\left[ R(\tau)\left( \sum_{j=0}^{H_\ell-1} \nabla_\theta^2 \log \pi_{\theta^K(\nu_t)}(a_j|s_j) \right) \right] \tag{118}$$

$$= \mathbb{E}_{P(\tau;\theta)}\mathbb{E}_\pi\left[ R(\tau)\left( \sum_{j=0}^{H_\ell-1} [\widehat{\phi}(s,a')\phi(s,a')^T - \phi(s,a)\phi(s,a')^T] \right) \right]$$

From the above, it is evident to ensure non-singular eigenvalues, we need to ensure non-singularity for the function approximation matrix $\phi\phi^T$ which has been a standard assumption in several settings (Sutton et al., 2009; Maei et al., 2009)

