# OpenReview forum: "Principal-Driven Reward Design and Agent Policy Alignment via Bilevel-RL"
_ICML.cc/2023/Workshop/ILHF — ILHF Workshop ICML 2023_

### Official Review · Reviewer_UQGw · 2023-06-03

**Rating:** 6
**Confidence:** 2

**Review:**

This paper addresses the challenge of aligning reinforcement learning (RL) agents with broader societal goals. The authors frame this as a bilevel optimization problem, with the outer level focused on designing rewards and the inner level aligning the policy with these designed rewards. To solve this, they introduce a new algorithmic framework known as Principal-driven Policy Alignment via Bilevel RL (PPA-BRL). This framework employs an iterative procedure to jointly solve design and policy parameters. They provide theoretical assurances that their methodology converges to a local optimal point. This work offers a new perspective on RL agent alignment, introducing a unique approach to policy design and alignment.

I think the paper formulates an interesting and meaningful problem. It also provides solid methodological and theoretical contributions. However, I was not quite sure if the paper is relevant to the main topic of the workshop, i.e., use of and learning with human feedback. Experiments are too simple as well, so it would be valuable to test the proposed algorithm for a rage of environment to really show its generalizability and applicability.

---

### Official Review · Reviewer_2UMD · 2023-06-18

**Rating:** 7
**Confidence:** 3

**Review:**

**Summary:**

The paper introduces PPABRL, a novel approach for reinforcement learning (RL) that addresses the limitations of traditional methods. By formulating the problem as a bilevel optimization framework within a principal-agent relationship, PPABRL aligns the agent's policy with the broader goals specified by the principal. It efficiently learns a reward parameterization that encompasses considerations like state space coverage, safety, and societal impacts. The paper provides a rigorous analysis of the framework, demonstrating convergence to a stationary point. Several examples, including energy-efficient manipulation tasks and cost-effective robotic navigation, highlight the merits of PPABRL in achieving policy alignment with diverse objectives. PPABRL offers a promising avenue for more effective and socially responsible RL decision-making.

**Strengths:**

1. The paper formulates the agent policy alignment problem as a bilevel optimization, where the outer objective focuses on reward design and the inner level pertains to policy alignment.
2. By employing differentiation and the Implicit function theorem to analyze local Karush-Kuhn-Tucker (KKT) points, the paper develops an iterative procedure called Principal-driven Policy Alignment via Bilevel RL (PPA-BRL). This procedure enables the joint optimization of design parameters at the outer level and policy parameters at the inner level within the framework.
3. Provides proofs to verify the proposed methodology converges to a local KKT point of the problem.

**Weaknesses:**

Despite the theoretical contributions of this paper. The numerical evaluation is not strong. I would like to see the broader impact of this method to real world RL alignment problems, such as training safer self-driving policies and human-like chatbots. Finetuning them is not expensive compared with training large models.

---

### Decision · Program_Chairs · 2023-06-20

Accept